# OntoFAR: Hierarchical Multi-Ontology Fusion Better Augments EHR Representation

## Abstract

Medical ontology graphs, which typically organize and relate comprehensive medical concepts in a hierarchical structure, are able to map a rich set of external knowledge onto the specific medical codes observed in electronic health records (EHRs). Through the connectivity in ontologies, healthcare predictive models can utilize the ancestor, descendant, or sibling information to add supplementary contexts to medical codes, thereby augmenting expressiveness of EHR representations. However, existing approaches are limited by the heterogeneous isolation of different ontology systems (e.g., conditions vs. drugs), that different types of ontology concepts have to be learned individually, and only the homogeneous ontology relationships can be exploited. This limitation restricts the existing methods from fully leveraging the cross-ontology relationships which could substantially enhance healthcare representations. In this paper, we propose OntoFAR, a framework that fuse multiple ontology graphs, utilizing the collaboration across ontologies to enhance medical concept representation. Our method jointly represents medical concepts cross multiple ontology structures by performing message passing in two dimensions: (1) vertical propagation over levels of ontology hierarchy, and (2) horizontal propagation over co-occurring concepts in EHR visits. Additionally, OntoFAR leverages the large language models (LLMs) pre-trained on massive open world information to understand each target concept with its ontology relationships, providing enhanced embedding initialization for concepts. Through extensive experimental studies on two public datasets, MIMIC-III and MIMIC-IV, we validate the superior performance of OntoFAR over the state-of-the-art baselines. Beyond accuracy, our model also exhibits the add-on compatibility to boost existing healthcare prediction models, and demonstrate a good robustness in scenarios with limited data availability.[1]

## 1 Introduction

With the ubiquity of electronic health records (EHRs) in modern healthcare systems, developing machine learning models to analyze comprehensive medical histories from large-scale patient populations has shown great potential in enhancing a wide range of predictive tasks (Choi et al., 2016a; Poulain & Beheshti, 2024; Jiang et al., 2023; Moghaddam et al., 2024). However, the inherent complexity of EHRs characterized by diverse, sparse, and temporal code appearance presents a challenge for learning expressive and robust medical concept representation. A promising solution to this challange is to integrate external medical ontology graphs. to enhance representation learning.

By providing a rich domain contexts for medical codes and their interrelationships, ontologies offer a rich source of knowledge bases to explain the medical code we observed in EHR. For instance, the ICD ontology classifies diseases based on symptoms, complaints, and causes, while the Anatomical Therapeutic Chemical (ATC) ontology categorizes drugs according to organ systems, ingredients, and functions. These ontologies provide hierarchical concepts that define specific codes, such as diabetes, with general to specific classification. By incorporating the related concepts in different hierarchies, the models can better capture the relevance between the medical code to improve the robustness of representations, especially in challenging scenarios such as learning rare disease. Therefore, recent studies have increasingly explored the augmentation of EHR representation through

---

[1]The implementation code is available at `https://anonymous.4open.science/r/OntoFAR-35D4`

the integration of supplementary ontology graphs (Choi et al., 2017; Shang et al., 2019; Ye et al., 2021; Zhang et al., 2020). However, a critical limitation in the current literature lies in the inability to accommodate multiple ontology systems in a unified learning framework. Specifically, existing methods typically focus on learning each ontology as an independent structure where only the vertical message passing (e.g., top-down concept aggregation) is facilitated. As a result, only the intra-ontology relationship linking the homogeneous concepts (e.g., disease-to-disease) is adopted for representing each medical code. To this end, there exists a need of fusing multiple ontologies and introducing the cross-ontology relationship (e.g., disease-to-drug, disease-to-procedure), so that the rich and diverse medical knowledge bases can be fully leveraged to augment EHR representations.

To address the gap of fusing diverse ontology graphs for augmenting EHR representation, we propose a multifaceted graph learning architecture, **OntoFAR** (Hierarchical **Onto**logy **F**usion for **A**ugmenting **R**epresentation), aiming to enable graph message passing across multiple ontologies in both vertical and horizontal dimensions. Specifically, OntoFAR facilitates (1) intra-ontology concept association through vertical propagation across hierarchical levels within each ontologies, and (2) inter-ontology concept fusion through horizontal propagation to connect co-occurring concepts in EHR data over all the ontology levels in parallel. By introducing the horizontal message passing as a new dimension, OntoFAR is advantageous in (1) connecting different ontologies at all levels of hierarchy, (2) capturing concept co-occurrence at all levels of EHR granularity, and (3) mining EHR patterns integrally with ontology structure learning. Furthermore, OntoFAR leverages the pretrained knowledge of large language models (LLMs) to initialize dense embeddings that can benefit from extensive open-world information. Last, our ontology representation framework can serve as a add-on component to most healthcare predictive models (e.g., RETAIN) for performance boosting and robustness enhancement.

To demonstrate the significance of our work, we conducted extensive experiments on two widely used EHR datasets, MIMIC-III and MIMIC-IV performing the task of sequential diagnosis prediction, including performance enhancement analysis when integrating to EHR models, baseline comparisons, data insufficiency tests, and interpretative case studies. The results demonstrate that OntoFAR, as a plug-in medical concept encoder, significantly improves the encoding phase of healthcare predictive models, leading to enhanced predictive healthcare performance.

## 2 RELATED WORK

**EHR Predictive Models**: The widespread adoption of electronic health records (EHRs) has facilitated the development of numerous machine learning models for predictive tasks in healthcare. Early efforts began with pioneering sequential models (Choi et al., 2016a; Ashfaq et al., 2019), followed by attention-based models (Choi et al., 2016b), and later transformer-based approaches (Li et al., 2020; Choi et al., 2020; Nayebi Kerdabadi et al., 2023). More recently, advanced structures like graph neural networks (GNNs) have been employed (Su et al., 2020; Lu et al., 2021b; Xu et al., 2022; Yang et al., 2023b; Poulain & Beheshti, 2024), further enhancing predictive capabilities.

**Ontology-augmented Medical Concept Representation Learning.** These works aim to enhance medical concept representation learning by augmenting structured EHR data with hierarchical medical ontology, without using any other data modalities (e.g. unstructured clinical text). For instance, GRAM (Choi et al., 2017) leverages the ontology hierarchy to represent a medical concept as a convex combination of itself and its ancestors. Building on GRAM,MMORE (Song et al., 2019) enhances GRAM by enabling multiple representations for each parent concept, addressing discrepancies between EHR data and medical ontologies. KAME (Ma et al., 2018) further improves prediction accuracy by incorporating ontology knowledge throughout the entire prediction process on top of code representation learning. Despite improved performance from GRAM-based methods, they only consider the unordered ancestors of a concept, limiting its expressibility by not fully leveraging the hierarchy. HAP (Zhang et al., 2020) overcomes this limitation by propagating attention hierarchically across the entire ontology, enabling a medical concept to adaptively learn its embedding from all concepts, not just its ancestors. ADORE (Cheong et al., 2023) utilizes the relational ontology SNOMED to integrate multi-source medical codes, whereas KAMPNet (An et al., 2023) employs contrastive learning to achieve effective EHR representation learning.

**Multi-modal Data Augmentation for Medical Concept Representation Learning.** GCL (Lu et al., 2021a) is a collaborative graph learning model that jointly learns patient and disease repre-

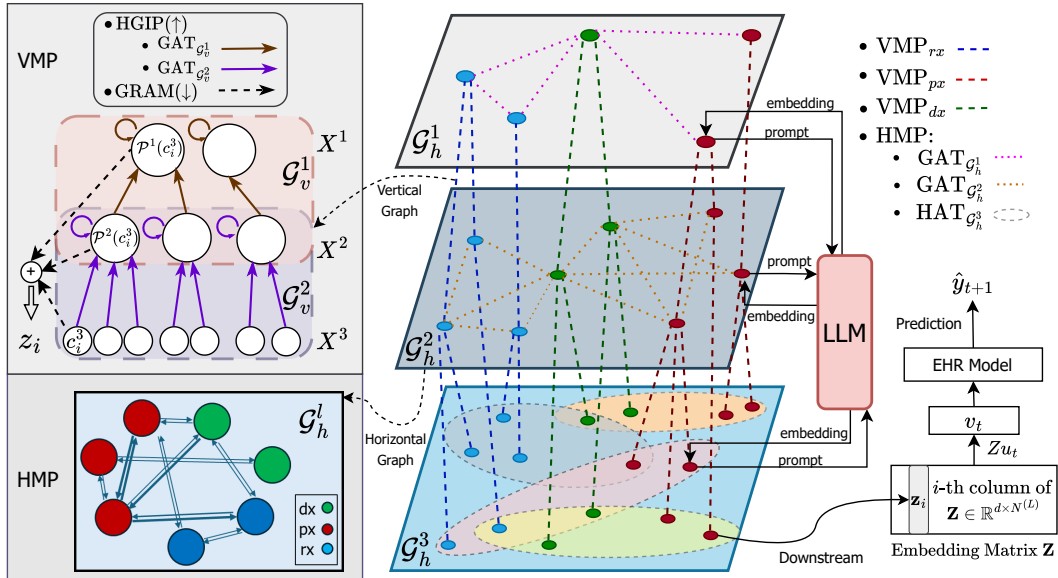

Figure 1: OntoFAR consists of three main steps: (1) Meta-KG construction, which hierarchically join the heterogeneous medical concepts from multiple ontology systems. Meta-KG has two dimensions: the vertical ontology graphs and the horizontal co-occurrence graphs. The initialization of concept embeddings is done through LLM prompting and dense retrieval. (2) Horizontal Massage Passing (HMP), which connect concept nodes across ontology graphs at different levels based on concept co-occurrence in EHR visits. HMP can be facilitated by either regular graph network (e.g., GAT) or hypergraph network (e.g., HAT). and (3) Vertical Massage Passing (VMP), where concept nodes in the same ontology graph are firstly propagated in bottom-up fashion through HGIP module, secondly propagated in top-down direction using GRAM module. The resulting embedding of medical codes at the lowest level are eventually used in downstream predictive tasks.

sentations, incorporating unstructured text with attention regulation. GRAPHCARE (Jiang et al., 2023) extracts information from LLMs and external biomedical knowledge graphs (KGs) to build patient-specific KGs. MedPath (Ye et al., 2021) enhances health risk prediction by incorporating personalized information by extracting a personalized knowledge graph (PKG) for each patient from SemMed web. RAM-EHR (Xu et al., 2024) improves EHR predictions by retrieving external textual knowledge for medical concepts from multiple online sources, augmenting the local model co-trained with consistency regularization.

## 3 METHODOLOGY

### 3.1 NOTATION AND PROBLEM DEFINITION

**EHR Dataset.** Denoted by $\mathcal{D} = \{\mathcal{X}_j \mid j \in \mathcal{J}\}$, an EHR dataset consists of the medical histories from a collection of patients $\mathcal{J}$. For each patient, his or her history $\mathcal{X}_j$ consists of a sequence of $T_j$ clinical visits thereby $\mathcal{X}_j = \{V_t\}_{t=1}^{T_j}$. Each visit $V_t$ is a set of $N_t$ medical code thereby $V_t = \{c_i\}_{i=1}^{N_t}$, where each code $c$ can indicate a diagnosis (*dx*), a prescription (*rx*), or a procedure (*px*). Each medical code $c$ can also be associated with a descriptive name $\mathcal{S}(c)$, which is typically a short text snippet provided by ontology.

**Medical Ontology.** A medical ontology is a hierarchical tree-like structure that organizes clinically related concepts from general categories at the upper levels, to specific code at the lower levels[2]. For different systems of medical concepts (e.g. diseases vs. drugs), there exists a separated ontology (ICD vs. ATC). A medical concept in a ontology is denoted as $c_i^{(l)} \in \boldsymbol{c}^{(l)}$, where $l \in [1 : L]$ indexes the ontology level (from the highest to lowest), $i \in [1 : N^{(l)}]$ indexes the concept at the $l$-th level, and $\boldsymbol{c}^{(l)}$ represents the set of all concepts at level $l$. Last, we define the function $\mathcal{P}^k(c_i^{(l)})$, which

---

[2]A medical code in EHRs is also a "concept" locates at leaf nodes in ontologies.

maps a concept $c_i^{(l)}$ at level $l$ to its ancestor or descendant concepts at level $k$. If $k > l$, it returns the set of descendant; if $k < l$, it returns the ancestor; and if $k = l$, it returns the concept itself.

**Sequential Diagnosis Predictive Task.** Given a patient's sequence of previous visits, $\mathcal{X}_j = \{V_1, V_2, \ldots, V_{T_j}\}$, the objective is to predict the diagnosis codes for the next visit $V_{T_j+1}$.

### 3.2 MODEL OVERVIEW

We present an overview of our proposed method, OntoFAR, a hierarchical multi-ontology fusing encoder of medical concepts designed to enhance the representation of EHR data. The entire framework depicted in Figure 1 is summarized in three key steps:

**Step 1:** Formulating the Meta-KG denoted by $\mathcal{G}$, a heterogeneous knowledge graph (KG) that joins multiple ontology graphs through each level of hierarchy. The initialization of node embeddings in Meta-KG is gained through LLM prompting and dense retrieval. This Meta-KG has $L$ horizontal inter-ontology parallel graphs at each level for horizontal massage passing, and two vertical inter-ontology graphs for each ontology, one bottom-up and one top-down, for vertical massage passing.

**Step 2:** Carrying out Horizontal Message Passing (HMP) over inter-ontology concept edges using EHR information. We use two options to construct horizontal graphs in Meta-KG: (1) regular graph structure where edges are defined based on the co-occurrence probabilities of concepts; and (2) hypergraph structure where each edge links all the concepts in a EHR visit.

**Step 3:** Performing Vertical Message Passing (VMP) over intra-ontology concept edges derived from parent-child relationships in ontology hierarchy. Within each ontology system, the process begins with a bottom-up propagation which passes up the embedding of each concept to its ancestor first, then concludes with top-down propagation where concept embeddings are passed down yielding the final embeddings for leaf nodes using GRAM (Choi et al., 2017).

### 3.3 STEP 1: META-KG INITIALIZATION WITH LLM

We use embedding $x_i^{(l)} \in \mathbb{R}^d$ to represent each medical concept $c_i^{(l)}$ in the Meta-KG. To initialize $x_i^{(l)}$ for each concept, we extract the concept name associated with concept code $c_i^{(l)}$, denoted by $\mathcal{S}(c_i^{(l)})$. Following this, we develop a prompting strategy specifically tailored for EHR description to retrieve embeddings from LLMs for each concept. Impirically, the best results came from the following prompting (Pr) strategy:

$$
\text{Pr} = \begin{cases} \text{``\{type\} code } \{c_i^{(l)}\} \text{ represents } \{\mathcal{S}(c_i^{(l)})\}, \text{ which is a general medical concept.''} & \text{if } l = 0 \\ \text{``\{type\} code } \{c_i^{(l)}\} \text{ represents } \{\mathcal{S}(c_i^{(l)})\}. \text{ It is a specific medical concept under} \\ \text{the categories of } \{\mathcal{P}^{l-1}(c_i^{(l)})\}(\{\mathcal{S}(\mathcal{P}^{l-1}(c_i^{(l)}))\}), \ldots, \{\mathcal{C}^1(c_i^{(l)})\}(\{\mathcal{S}(\mathcal{P}^1(c_i^{(l)}))\}).\text{''} & \text{if } l > 0 \end{cases}
$$

where "type" refers to name of the ontology concept system (e.g. ICD9 diagnosis/procedure, and ATC drug). Therefore, for each code, we first provide a descriptive text for the code itself and then mention its broader concept categories or EHR ancestors by locating them at higher levels of the ontology using the mapping function $\mathcal{P}$. The code type can be a diagnosis, prescription, or procedure. An example of an LLM prompt for a diagnosis code is as follows:

**ICD9 Diagnosis 250.7:** *Prompt: "ICD9 diagnosis code 250.7 represents Diabetes with peripheral circulatory disorders. It is a specific medical concept under the broader categories of 250 (Diabetes mellitus), 249-259 (Diseases of Other Endocrine Glands), and 240-279 (Endocrine, Nutritional, and Metabolic Diseases, and Immunity Disorders)."*

We employ the OpenAI off-the-shelf LLMs, GPT text-embedding-3-small/text-embedding-3-large (OpenAI, 2023), denoted as $\mathcal{LLM}$ to generate a semantic embedding, containing clinical knowledge and context background from LLMs. We initialize the vector representation $x_i^{(l)} \in \mathbb{R}^d$ as follows:

$$
x_i^{(l)} = \mathcal{LLM}(\text{Pr}(c_i^{(l)})), \quad l = 1, \ldots, L, \quad i = 1, \ldots, N^{(l)} \tag{1}
$$

### 3.4 STEP 2: INTER-ONTOLOGY HORIZONTAL MESSAGE PASSING (HMP)

Given Meta-KG, we aim to learn horizontal graphs where medical concepts are connected within or across different ontologies at all levels. Through the co-occurrence relationship observed from

EHR visits, this step will utilize the edges that connected the concepts appeared in the same visits and perform graph message passing using GNNs. Horizontal graph edges at the higher levels of hierarchy will be created based on the ancestor concepts that are mapped from the observed codes at the lowest level. This operation will achieve two key goals: 1) fusing diagnosis, drug, and procedure ontologies to capture heterogeneous code interactions, and 2) utilizing information at all levels of granularity for EHR representation. While leaf-level EHR codes offer detailed but sparse insights, mapping to higher-level concepts reduces the graph complexity, allowing us to leverage both fine-grained and coarse-grained information simultaneously for representation learning.

To construct graph edges based on co-occurrence information, we first create a leaf (child) level co-occurrence count matrix $Q^{(L)} \in \mathbb{R}^{N^{(L)} \times N^{(L)}}$, where $Q_{ij}$ denotes the number of occurrences of leaf code $c_j^{(L)}$ given the presence of leaf code $c_i^{(L)}$ within a hospital visit. We then derive the co-occurrence matrices at higher levels by aggregating the co-occurrence counts of their children as follows:

$$Q_{pq}^{(l)} = \sum_{c_i^{(L)} \in \mathcal{C}(p)} \sum_{c_j^{(L)} \in \mathcal{C}(q)} Q_{ij}, \quad l = 1, \ldots, L \tag{2}$$

where $\mathcal{C}(p) = \mathcal{P}^L(c_p^{(l)})$ and $\mathcal{C}(q) = \mathcal{P}^L(c_q^{(l)})$ denote the sets of leaf-level children of parent-level nodes $c_p^{(L)}$ and $c_q^{(L)}$, respectively. Next, we derive the co-occurrence conditional probability matrix $P^{(l)}$ from the count matrix $Q^{(l)}$ by normalizing each entry with the total occurrences of the corresponding node $p$, expressed as: $P_{pq}^{(l)} = Q_{pq}^{(l)} / \sum_j Q_{pj}^{(l)}$, for $l = 1 : L$. The co-occurrence probability matrix is then used to define edges in the graph. Specifically, an edge between nodes $p$ and $q$ is included if the co-occurrence probability exceeds a threshold $\tau^{(l)}$. This binarization generates the adjacency matrix $\mathcal{A}_h^{(l)}$ from $P^{(l)}$ as: $\mathcal{A}_h^{(l)} = \mathbb{I}(P^{(l)} \geq \tau)$, where $\mathbb{I}(\cdot)$ is the indicator function, returning 1 if true and 0 otherwise. Consequently, we construct the horizontal graphs $\mathcal{G}_h^{(l)} = (\mathcal{V}^{(l)}, \mathcal{E}^{(l)})$, where $\mathcal{V}^{(l)} = \mathbf{c}^{(l)}$ and $\mathcal{E}^{(l)} = \mathcal{A}_h^{(l)}$, with $N^{(l)}$ nodes and $M^{(l)}$ edges at the $l$-th level of the ontology. Note that since $P_{pq}^{(l)} \neq P_{qp}^{(l)}$ in general, $\mathcal{A}_h^{(l)}$ is not necessarily a symmetric matrix. Therefore, $\mathcal{G}_h^{(l)}$ represents a directed graph. We employ a regular graph neural operator, such as GAT (Veličković et al., 2017) leveraging the multihead attention mechanism, to encode medical codes in each level:

$$\mathbf{X}_{(k+1)}^{(l)} = \text{GAT}\left(\mathbf{X}_{(k)}^{(l)}, \mathcal{A}_h^{(l)}\right), \quad l = 1, \ldots, L \tag{3}$$

where $\mathbf{X}_{(k+1)}^{(l)} \in \mathbb{R}^{N^{(l)} \times d}$ and $\mathbf{X}_{(k)}^{(l)} \in \mathbb{R}^{N^{(l)} \times d}$ denote the node features at the $(k+1)$-th and $k$-th layers of $\mathcal{G}_h^{(l)}$, respectively. Alternatively, for the leaf level of the ontology ($l = L$), we can utilize a hypergraph structure due to its robust capability to capture the high-order complex relationships between visits and medical codes (Xu et al., 2023; 2024; Cai et al., 2022). In this approach, visits are treated as hyperedges $\mathcal{E}^{(L)} = V$, and leaf-level medical codes are treated as nodes $\mathcal{V}^{(L)} = \mathbf{c}^{(L)}$. This allows us to construct $\mathcal{G}_h^{(L)} = (\mathcal{V}^{(L)}, \mathcal{E}^{(L)})$ at the leaf level of the ontology with $N^{(L)}$ nodes and $M^{(L)}$ hyperedges. We employ the Hypergraph Attention Network (HAT) (Bai et al., 2021) to encode this leaf-level hypergraph:

$$\mathbf{X}_{(k+1)}^{(L)} = \text{HAT}\left(\mathbf{X}_{(k)}^{(L)}, \mathcal{H}^{(L)}\right) \tag{4}$$

Here, $\mathbf{X}_{(k+1)}^{(L)} \in \mathbb{R}^{N^{(L)} \times d}$ and $\mathbf{X}_{(k)}^{(L)} \in \mathbb{R}^{N^{(L)} \times d}$ denote the node features in the $(k+1)$-th and $k$-th layers of $\mathcal{G}_h^{(L)}$, respectively. $\mathbf{H}_h^{(L)} \in \mathbb{R}^{N^{(L)} \times M^{(L)}}$ is the incidence matrix mapping nodes to edges.

We avoid using hypergraphs for the ancestral levels, which involve fewer nodes. Employing hypergraphs at these levels would necessitate defining hyperedges with visits, resulting in an excessive number of hyperedges relative to the fewer nodes. This leads to a dense graph where nodes are redundantly connected, causing embeddings to become overly similar, despite representing distinct entities. Instead, we employ a regular graph structure, allowing for one-to-one edges based on co-occurrence, with edge inclusion controlled by a predefined co-occurrence probability threshold.

## 3.5 STEP 3: INTRA-ONTOLOGY VERTICAL MESSAGE PASSING (VMP)

Also in Meta-KG, we aim to utilize the vertical graphs which are the hierarchies from each individual ontology. The message passing over ancestor to descendant levels, enables the information sharing

across ontology concepts at different levels of granularity. Inspired by ideas of attention propagation in (Zhang et al., 2020) and the "two-round propagation" approach in the belief propagation algorithm (Pearl, 2022), we design the VHG module, a two-round progressive graph-based encoding technique that can integrate information across all ontology levels.

First round, the bottom-up propagation, referred to as Hierarchical Graph Information Propagation (HGIP), adaptively updates each concept node in the ontology as a convex combination of itself and its child concepts using multi-head attention mechanism. This process begins by constructing a series of directed vertical subgraphs consisting of a pair of adjacent ontology levels, starting with $\mathcal{G}_v^{(L-1)}$ (connecting level $L$ to $L-1$) and continuing up to $\mathcal{G}_v^{(1)}$ (connecting level 2 to level 1). Edges in each subgraph are defined by parent-child relationships, with directed edges from nodes in level $l$ to their parent nodes in level $l-1$, forming the adjacency matrix $\mathcal{A}_v^{(l)}$ for the vertical subgraph $\mathcal{G}_v^{(l)}$. We then apply a graph attention operator, such as GAT, to each subgraph, encoding the hierarchical structure in a bottom-up manner. Starting with $\mathcal{G}_v^{(L-1)}$, parent node embeddings in level $L-1$ are updated by aggregating information from their children in level $L$ using multi-head attention. These updated results are then incorporated into $\mathcal{G}_v^{(L-2)}$, where the child embeddings are the updated parent nodes from $\mathcal{G}_v^{(L-1)}$. This sequential process continues to the root level subgraph $\mathcal{G}_v^{(1)}$, propagating information throughout the ontology. HGIP is formally expressed as:

$$\text{For } s = 0, \ldots, L-1: \quad [\mathbf{X}_{(k+1)}^{(L-s)}, \mathbf{X}_{(k+1)}^{(L-s+1)}] = \text{GAT}_{\mathcal{G}_v^{(L-s)}}\left([\mathbf{X}_{(k)}^{(L-s)}, \mathbf{X}_{(k)}^{(L-s+1)}], \mathcal{A}_v^{(L-s)}\right) \quad (5)$$

where $\mathbf{X}_{(k)}^{(l-s)} \in \mathbb{R}^{N^{(l-s)} \times d}$ and $\mathbf{X}_{(k)}^{(l-s+1)} \in \mathbb{R}^{N^{(l-s+1)} \times d}$ represent the embeddings of parent and child nodes at the $(l-s)$-th and $(l-s+1)$-th level of the ontology in the layer of $\mathcal{G}_v^{(l-s)}$.

Instead of defining a single graph for the entire ontology, we represent it as a series of sequential subgraphs, each corresponding to a pair of adjacent levels. This approach firstly ensures that each parent node in the hierarchy contains the curated distilled information of all its descendants in lower levels. Second, it allows us to effectively control the order of node embedding updates across the main graph. This enhances the bottom-up HAP method (Zhang et al., 2020) in two ways: 1) it employs a sequential GNN structures for efficient, parallel node updates, and 2) it integrates a multi-head attention mechanism to compute attention weights, enabling expressive multi-view representations and addressing inconsistencies between EHR co-occurrences and ontologies (Song et al., 2019).

Second round, we apply GRAM (Choi et al., 2017) to compute the final representation of the leaf-level nodes by adaptively aggregating information from their ancestors using attention mechanism. The final representation $z_i \in \mathbb{R}^{d_c}$ of each leaf-level code $c_i^{(L)}$, where $i = 1 : N^{(L)}$, is computed as a convex combination of child embedding $x_i^{(L)}$ and all its ancestors' embeddings:

$$\text{GRAM}: \quad z_i = \sum_{l=1}^{L} \alpha_{il} \mathcal{P}^l(x_i^{(L)}), \qquad \alpha_{il} \geq 0, \quad \text{for} \quad l = 1, \ldots, L \quad (6)$$

where $\alpha_{il} \in \mathbb{R}^+$ denotes the attention weight for the code embedding $\mathcal{P}^l(x_i^{(L)})$ in computing $z_i$. The attention weight $\alpha_{il}$ is computed using the softmax function as:

$$\alpha_{il} = \frac{\exp(f(x_i^{(L)}, \mathcal{P}^l(x_i^{(L)})))}{\sum_{k=1}^{L} \exp(f(x_i^{(L)}, \mathcal{P}^k(x_i^{(L)})))} \quad (7)$$

where $f(a, b)$ is a multi-layer perceptron that produces a scalar energy representing the raw attention between $a$ and $b$. The softmax function normalizes the energies into attentions between 0 and 1.

### 3.6 Enhancing Downstream Task with OntoFAR

We introduce OntoFAR as a complementary medical concept representation learning module. The final concept representations produced by OntoFAR, $\mathbf{z}_1, \mathbf{z}_2, \ldots, \mathbf{z}_{N^{(L)}}$, are concatenated to form the embedding matrix $\mathbf{Z} \in \mathbb{R}^{d \times N^{(L)}}$, where $\mathbf{z}_i$ is the $i$-th column of $\mathbf{Z}$. This embedding matrix will be used in a downstream task, such as diagnosis prediction. Formally, for sequential diagnosis

prediction we have $f : (\{V_1, V_2, \ldots, V_t\}) \rightarrow \hat{\mathbf{y}}_{t+1}$, where $\hat{\mathbf{y}}_{t+1} \in \mathbb{R}^{N_{dx}^{(L)}}$ is a multi-hot vector, with $N_{dx}^{(L)}$ denoting the total number of diagnosis codes:

$$
\begin{aligned}
\mathbf{Z} = [\mathbf{z}_1, \mathbf{z}_2, \ldots, \mathbf{z}_{N^{(L)}}] &\leftarrow \text{OntoFAR}(\mathbf{x}_1^{(L)}, \mathbf{x}_2^{(L)}, \ldots, \mathbf{x}_{N^{(L)}}^{(L)}) \\
\mathbf{v}_1, \mathbf{v}_2, \ldots, \mathbf{v}_t &= \mathbf{Z}[\mathbf{u}_1, \mathbf{u}_2, \ldots, \mathbf{u}_t] \\
\mathbf{h}_1, \mathbf{h}_2, \ldots, \mathbf{h}_t &= \text{Encoder}(\mathbf{v}_1, \mathbf{v}_2, \ldots, \mathbf{v}_t) \\
\mathbf{h}_p &= \text{Aggregate}(\mathbf{h}_1, \mathbf{h}_2, \ldots, \mathbf{h}_t) \\
\hat{\mathbf{y}}_{t+1} &= \text{Sigmoid}(\mathbf{W}\mathbf{h}_p + \mathbf{b})
\end{aligned}
\tag{8}
$$

For each visit $V_t$, we obtain a representation $\mathbf{v}_t \in \mathbb{R}^d$ by multiplying the final embedding matrix $\mathbf{Z}$ with a multi-hot vector $\mathbf{u}_t = \{0, 1\}^{N^{(L)}}$, which represents clinical events existence in the visit. The sequence of visit representations $\{\mathbf{v}_1, \mathbf{v}_2, \ldots, \mathbf{v}_t\}$ serves as input to a main encoder, $\text{Encoder}(\cdot)$, such as a Transformer or Retain, producing the encoded hidden embedding $\mathbf{h}_t$ for the $t$-th visit. The patient representation $\mathbf{h}_p \in \mathbb{R}^d$ is derived by aggregating visit embeddings using a function, $\text{Aggregate}(\cdot)$, which may be summation, averaging, or attention-pooling. The final prediction is computed by applying a Sigmoid function to the linear transformation of $\mathbf{h}_p$, with $\mathbf{W} \in \mathbb{R}^{N^{(L)} \times d}$ and $\mathbf{b} \in \mathbb{R}^{N^{(L)}}$ as the weight and bias, respectively. The output $\hat{\mathbf{y}}_{t+1}$ is the predicted vector in $\mathbb{R}^{N^{(L)}}$. The loss at each timestamp is calculated using cross-entropy between the ground truth $y_{t+1}$ and predicted visit $\hat{y}_{t+1}$. Algorithm 1 in Appendix A.1 outlines the OntoFAR process.

## 4 EXPERIMENTAL SETTING

**Datasets:** We utilize two publicly available datasets: MIMIC-III (Johnson et al., 2016) and MIMIC-IV (Johnson et al., 2023). MIMIC-III (2001–2012) uses ICD-9 codes, while MIMIC-IV (2008–2019) includes both ICD-9 and ICD-10 and provides more comprehensive longitudinal data. Prescription codes in both datasets follow the National Drug Code (NDC) system, which we map to the Anatomical Therapeutic Chemical (ATC) Classification. Table 1 presents the cohort statistics. This task is predicting ICD-9 diagnosis codes for the next visit (4,283 unique codes in MIMIC-III, 8,818 in MIMIC-IV). We present experiments on mortality and readmission prediction in Appendix A.2.

Table 1: Data statistics for MIMIC-III and MIMIC-IV

| Metric | MIMIC-III | MIMIC-IV | Metric | MIMIC-III | MIMIC-IV |
|---|---|---|---|---|---|
| # Samples | 12,430 | 25,028 | Conditions/sample | 29.02 | 66.84 |
| # Patients | 7,515 | 18,829 | Drugs/sample | 70.10 | 118.17 |
| # Visits | 12,430 | 25,028 | Unique drugs | 471 | 510 |
| # Labels/ sample | 13.32 | 11.89 | Procedures/sample | 7.01 | 5.77 |
| # Unique conditions (ICD) | 4,283 | 7,054 | Unique procedures | 1328 | 2033 |

**Implementations:** We report the mean and confidence intervals of the results after 5-fold cross-validation experimentation. OntoFAR uses 4 attention heads for horizontal graphs, 2 for vertical graphs, dropout rates of 0.1 and 0.2, respectively, and a shared embedding dimension of $d = 128$ for all nodes in the Meta-KG. We use a 3-level hierarchy ($L = 3$) for the ICD-9 diagnosis, ICD-9 prescription, and ATC drug ontologies in our experiments. Our implementation is compatible with PyHealth (Yang et al., 2023a).

**Evaluation Metrics:** (1) **AUPRC**: Measures the area under the precision-recall curve, reflecting the trade-off between precision and recall across different thresholds. (2) **Acc@k**: The number of correct diagnosis codes among the top $k$ predictions divided by $\mathbf{min}(k, \|y_t\|)$ where $\|y_t\|$ is the number of labels in the $(t+1)$-th visit. (3) **AUROC**: Measures the area under the receiver operating characteristic curve, which captures the trade-off between true positive and false positive rates. (4) **F1-score**: The harmonic mean of precision and recall, providing a balance between the two.

## 5 EVALUATION RESULTS

We investigate the following research questions: : **RQ1** Does our method improve various EHR models when added as a medical concept encoder? **RQ2** How does OntoFAR compare to existing

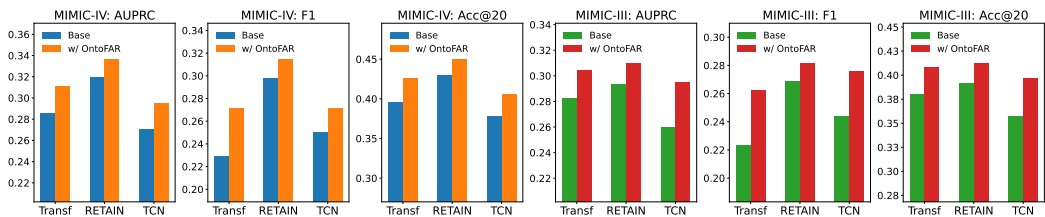

Figure 2: Performance enhancement evaluation before and after integrating OntoFAR into three diagnosis prediction models, using the MIMIC-III and MIMIC-IV datasets.

medical code encoders? **RQ3** What is the impact of each OntoFAR's component on performance? **RQ4** How does our method can alleviate the limitations of data insufficiency?

### 5.1 RQ1: PERFORMANCE ENHANCEMENT ANALYSIS

We propose that incorporating our medical concept encoder, OntoFAR, into existing EHR machine learning predictors to boost downstream performance through concept representation enhancement. To verify this, we integrated OntoFAR into three various predictive models: (1) **Transformer** (Vaswani, 2017), which leverages the power of the self-attention mechanism; (2) **RETAIN** (Choi et al., 2016b), an RNN-based model for EHRs that utilizes a two-level reverse time attention mechanism; and (3) **TCN** (Bai et al., 2018), a temporal convolutional network that uses causal convolutions to capture temporal dependencies in sequential data. We conducted experiments with each model both with and without OntoFAR. Figure 2 presents grouped bar plots that illustrate the comparative results, demonstrating that OntoFAR consistently improves predictive accuracy across all four models, validating its effectiveness in learning superior medical concept representations.

### 5.2 RQ2: BASELINE COMPARISON

We compare our method with four existing medical ontology structure encoders: **GRAM** (Choi et al., 2017), **MMORE** (Song et al., 2019), **KAME** (Ma et al., 2018), and **HAP** (Zhang et al., 2020). These encoders are designed for ontology-based augmentation of EHR representation and can also be added to predictive models as an extension to boost performance. We used the Transformer as the main diagnosis prediction model and tested five setups: (1) the main model without any medical concept encoder; (2) the main model with each of the four existing encoders; and (3) the main model with OntoFAR. We then compare the performance of each setup. General performance section in table 2 shows that OntoFAR outperforms the existing encoders in enhancing the predictive performance of the Transformer, demonstrating its effectiveness as a complementary medical concept encoder. We also test two different graph encoding techniques for $\mathcal{G}_h^{(L)}$. Both techniques outperformed baselines: HAT excelled on MIMIC-IV with a larger search space (8,818 codes), while GAT performed best on MIMIC-III with a narrower search space (4,283 codes).

Table 2: Performance comparison on MIMIC-III and MIMIC-IV based on PR-AUC, F1 score, and Acc@20. The reported values include means and 95% confidence intervals.

| D | Model | General Performance | | | Label Category Performance (AUPRC) | | | |
|---|---|---|---|---|---|---|---|---|
| | | **PR-AUC** | **F1** | **Acc@20** | **0-25%** | **25-50%** | **50-75%** | **75-100%** |
| MIMIC-IV | Transformer | $28.83_{\pm0.35}$ | $22.87_{\pm0.25}$ | $39.83_{\pm0.60}$ | $28.95_{\pm0.38}$ | $52.22_{\pm0.29}$ | $56.23_{\pm1.5}$ | $67.99_{\pm1.6}$ |
| | GRAM | $29.96_{\pm0.45}$ | $24.31_{\pm0.09}$ | $41.24_{\pm0.46}$ | $30.11_{\pm0.46}$ | $53.12_{\pm0.32}$ | $55.51_{\pm2.3}$ | $68.99_{\pm0.27}$ |
| | MMORE | $30.06_{\pm0.25}$ | $25.11_{\pm1.60}$ | $41.46_{\pm0.32}$ | $30.15_{\pm0.30}$ | $53.37_{\pm0.55}$ | $57.97_{\pm1.09}$ | $68.07_{\pm1.3}$ |
| | KAME | $29.13_{\pm0.32}$ | $23.39_{\pm0.32}$ | $40.28_{\pm0.32}$ | $28.84_{\pm0.32}$ | $52.22_{\pm0.32}$ | $55.66_{\pm0.32}$ | $68.08_{\pm0.32}$ |
| | HAP | $30.01_{\pm0.23}$ | $23.38_{\pm1.30}$ | $41.40_{\pm0.34}$ | $30.09_{\pm0.26}$ | $53.58_{\pm1.0}$ | $58.32_{\pm2.4}$ | $70.30_{\pm1.0}$ |
| | OntoFAR$_{w/ GAT}$ | $30.97_{\pm0.09}$ | $26.83_{\pm0.09}$ | $\mathbf{42.89_{\pm0.07}}$ | $31.03_{\pm0.02}$ | $55.11_{\pm0.97}$ | $58.79_{\pm0.86}$ | $\mathbf{71.35_{\pm0.29}}$ |
| | OntoFAR$_{w/ HAT}$ | $\mathbf{31.14_{\pm0.79}}$ | $\mathbf{27.11_{\pm0.06}}$ | $42.60_{\pm0.79}$ | $\mathbf{31.10_{\pm0.81}}$ | $55.86_{\pm1.0}$ | $\mathbf{59.62_{\pm0.49}}$ | $69.88_{\pm0.29}$ |
| MIMIC-III | Transformer | $28.23_{\pm0.24}$ | $22.36_{\pm0.33}$ | $38.03_{\pm0.34}$ | $28.07_{\pm0.38}$ | $54.20_{\pm0.29}$ | $50.62_{\pm1.5}$ | $74.14_{\pm0.10}$ |
| | GRAM | $28.99_{\pm0.34}$ | $23.62_{\pm0.48}$ | $39.14_{\pm0.52}$ | $28.84_{\pm0.41}$ | $54.58_{\pm1.3}$ | $50.58_{\pm0.34}$ | $74.44_{\pm0.77}$ |
| | MMORE | $29.11_{\pm0.38}$ | $23.67_{\pm0.70}$ | $39.14_{\pm0.53}$ | $28.87_{\pm0.43}$ | $54.92_{\pm0.87}$ | $51.29_{\pm0.25}$ | $74.42_{\pm0.25}$ |
| | KAME | $28.52_{\pm0.32}$ | $23.18_{\pm0.09}$ | $38.36_{\pm0.45}$ | $28.19_{\pm0.38}$ | $55.13_{\pm0.16}$ | $50.09_{\pm0.32}$ | $73.79_{\pm0.23}$ |
| | HAP | $29.28_{\pm0.41}$ | $23.25_{\pm0.88}$ | $39.46_{\pm0.55}$ | $29.10_{\pm0.45}$ | $55.11_{\pm0.99}$ | $52.19_{\pm0.27}$ | $76.28_{\pm0.24}$ |
| | OntoFAR$_{w/ GAT}$ | $\mathbf{30.43_{\pm0.37}}$ | $\mathbf{26.25_{\pm0.30}}$ | $\mathbf{40.80_{\pm0.40}}$ | $\mathbf{30.18_{\pm0.03}}$ | $\mathbf{56.23_{\pm0.03}}$ | $52.93_{\pm0.04}$ | $\mathbf{76.97_{\pm0.05}}$ |
| | OntoFAR$_{w/ HAT}$ | $30.27_{\pm0.38}$ | $26.05_{\pm1.00}$ | $40.52_{\pm0.54}$ | $30.08_{\pm0.46}$ | $55.67_{\pm0.14}$ | $\mathbf{53.22_{\pm0.3}}$ | $76.64_{\pm0.23}$ |

Table 3: Ablation study of OntoFAR using MIMIC-III and MIMIC-IV datasets. The reported values include means and 95% confidence intervals.

| Model | MIMIC-III | | | MIMIC-IV | | |
|---|---|---|---|---|---|---|
| | PR-AUC | F1 | Acc@20 | PR-AUC | F1 | Acc@20 |
| w/o HMP | $29.33_{\pm 0.33}$ | $24.99_{\pm 0.68}$ | $39.47_{\pm 0.44}$ | $30.30_{\pm 0.41}$ | $26.04_{\pm 1.00}$ | $41.78_{\pm 0.54}$ |
| w/o HGIP | $29.83_{\pm 0.43}$ | $24.77_{\pm 0.47}$ | $40.20_{\pm 0.39}$ | $29.71_{\pm 0.33}$ | $25.13_{\pm 0.44}$ | $41.38_{\pm 0.45}$ |
| w/o LLM | $29.43_{\pm 0.31}$ | $24.68_{\pm 0.52}$ | $39.43_{\pm 0.38}$ | $30.20_{\pm 0.40}$ | $25.18_{\pm 0.70}$ | $41.36_{\pm 0.42}$ |
| OntoFAR | $\mathbf{30.43_{\pm 0.37}}$ | $\mathbf{26.25_{\pm 0.30}}$ | $\mathbf{40.80_{\pm 0.40}}$ | $\mathbf{31.14_{\pm 0.79}}$ | $\mathbf{27.11_{\pm 0.06}}$ | $\mathbf{42.60_{\pm 0.79}}$ |

## 5.3 RQ3: ABLATION STUDY

As shown in Table 3, we conduct an ablation study to evaluate OntoFAR by removing key components: (1) **w/o HMP**: no horizontal message passing, (2) **w/o HGIP**: no hierarchical graph information propagation in vertical message passing, and (3) **w/o LLM**: no LLM for concept embedding initialization. All ablated versions showed performance drops, with **w/o HMP** removing ontology fusion, **w/o HGIP** weakening the use of hierarchical relationships for infomation sharing, and **w/o LLM** reducing domain knowledge integration. These results highlight the importance of each component in boosting model performance.

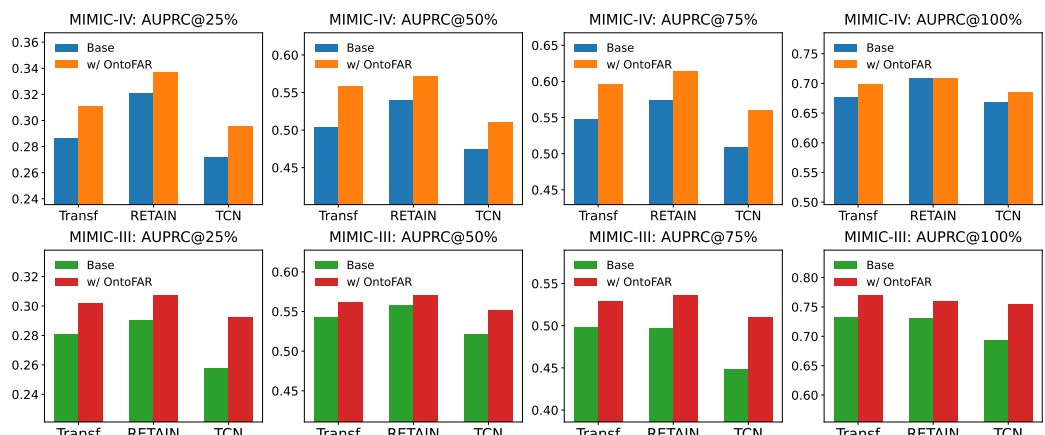

Figure 3: Performance evaluation across code frequency categories before and after integrating OntoFAR to three diagnosis prediction models, using the MIMIC-III and MIMIC-IV datasets.

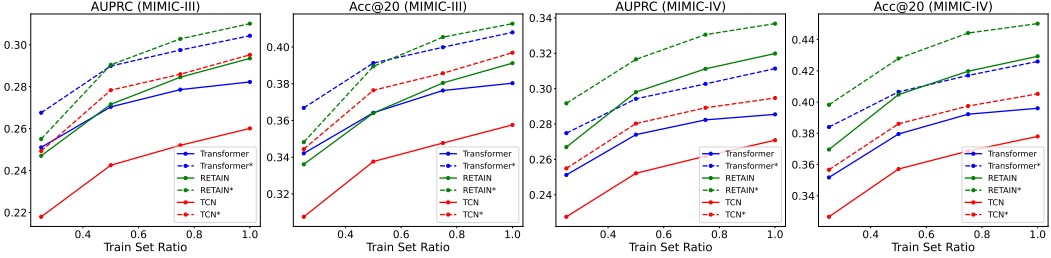

Figure 4: Performance evaluation across different training set sizes using the MIMIC-III and MIMIC-IV datasets. An asterisk (*) next to each encoder indicates the integration of OntoFAR.

## 5.4 RQ4: DATA INSUFFICIENCY ANALYSIS

To assess our model's robustness under data insufficiency, we conducted two experiments:

**Experiment 1: Performance on Predicting Rare Medical Codes.** We sort all diagnosis labels in the training set by frequency and divid them into four groups: 0-25%, 25-50%, 50-75%, and 75-100% percentiles, where the 0-25% group represents the rarest medical codes and the 75-100% group represents the most common. These varying frequencies can reflect different levels of data insufficiency. To evaluate our model's effectiveness in predicting rare medical codes, we compare its

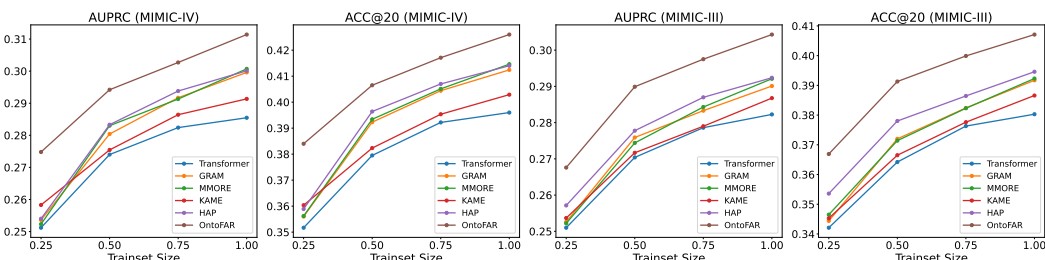

Figure 5: Performance comparison of baseline integration into the Transformer model across different training set sizes using the MIMIC-III and MIMIC-IV datasets.

performance across these groups. Figure 3 illustrates the performance improvement of integrating OntoFAR into each model based on PRAUC, revealing a substantial boost, especially for rare codes.

Right side of Table 2 further compares our method against other medical concept encoder baselines for rare code prediction, showing that OntoFAR consistently outperforms, particularly when data is insufficient for learning robust representations.

**Experiment 2: Varying Training Data Size.** In the second experiment, we vary the size of the training dataset to evaluate the model's performance under limited data conditions. Figure 4 shows that even with reduced training data, our model still improves the performance of EHR models significantly. Additionally, Figure 5 demonstrates that our model consistently outperforms its components in data-scarce scenarios, confirming its superiority and effectiveness.

### 5.5 CASE STUDY ANALYSIS

Figure 6 illustrates how OntoFAR learns the representation of the ICD-9 code 428.0, which denotes "Congestive heart failure, unspecified" (CHF), through a two-dimensional massage passing paradigm for rich medical concept representations. Vertically, OntoFAR retrieves all ancestors of this code across levels, and horizontally, it gathers co-occurring codes (red for procedures, green for diagnosis, and blue for drugs) for each

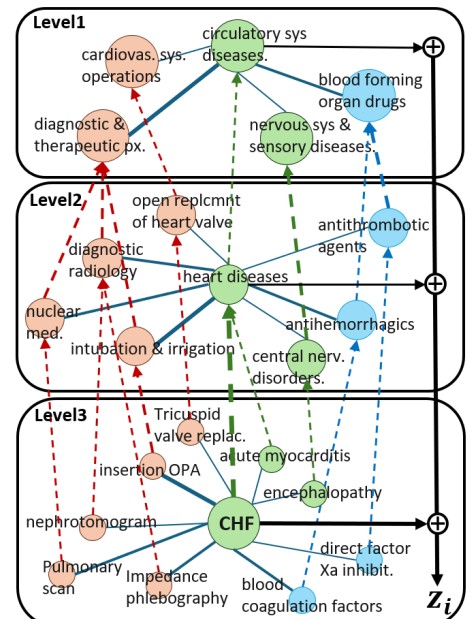

Figure 6: Case study: An example demonstrating how OntoFAR learns representation for a ICD-9 medical concept 428.0 representing "Congestive heart failure" or "CHF".

parent and the target code across all ontologies. The figure shows the extracted sub-KG for ICD-9 code 428.0 within the Meta-KG. Representation learning begins by initializing each node using LLM prompting and embedding retrieval. OntoFAR then performs horizontal propagation, applying graph attention to aggregate information from neighboring nodes across all levels. Next, the HGIP propagates information upward, updating each parent node using its children's embeddings via graph attention. Finally, the node embedding is refined through a convex combination of its own representation and those of its ancestors. The weights for all graph edges during horizontal and vertical propagation are learned through attention techniques, with edge thickness in the figure indicating the relative attention assigned to each edge.

## 6 CONCLUSION

We introduced OntoFAR, a multi-ontology fusion framework to augment medical concept representation in EHR models. OntoFAR extracts cross-ontology relationships through message passing in two dimensions: vertical and horizontal, and initializes concept embeddings with LLM prompting and dense retrieval. The proposed framework improves EHR prediction accuracy over state-of-the-art methods. Additionally, we showcase the robustness of OntoFAR in data-limited scenarios and validate its add-on compatibility to enhance existing healthcare models.

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

# A APPENDIX

## A.1 ONTOFAR OPTIMIZATION PROCESS

Algorithm 1 presents the OntoFAR training process. Though it assumes stochastic gradient updates for clarity, it can be readily extended to other gradient-based optimization methods, such as mini-batch.

---

**Algorithm 1** OntoFAR Optimization

---

1: **LLM Embedding Retrieval for Meta-KG Initialization:** Initialize the node embeddings inside Meta-KG with LLM prompting using Eq. 1
2: **Meta-KG Edge Construction:** (1) For HPG construct edges based on co-occurrence information. (2) For VHG construct edges using parent-child relationships.
3: **repeat**
4:     $\mathcal{X}_j \leftarrow$ random patient from dataset $\mathcal{D}$
5:     **for** visit $V_t$ in $\mathcal{X}_j$ **do**
6:         **for** code $c_i$ in $V_t$ **do**
7:             Refer to $\mathcal{G}$ to find its ancestors in all ontology levels ($l = 1 : L$) and their neighboring nodes
8:             **Horizontal Massage Passing (HMP):** Update the embeddings of the nodes and its ancestors in each horizontal graph $\mathcal{G}_h^{(l)}$ by aggregating neighboring nodes or hyperedges (Eq. 4, Eq. 3).
9:             **Vertical Massage Passing (VMP):** Use the HGIP module to (1) propagate information from the child node to its parents using the chain of sequential subgraphs $\mathcal{G}_v^l$ (Eq. 5). (2) derive the final node representation $z_i$ using GRAM (Eq. 6).
10:         **end for**
11:         Utilize final code representations to perform the downstream task (Eq. 8)
12:     **end for**
13:     Calculate the prediction loss and update the network parameters
14: **until** convergence

---

## A.2 COMPLEMENTARY EXPERIMENTAL RESULTS

Table 4 presents a performance comparison of OntoFAR with baseline models for the tasks of mortality prediction and readmission prediction using the MIMIC-III dataset. Additionally, Table 5 provides a comparison of baseline models for the diagnosis prediction task, incorporating a broader set of baselines compared to Table 2, also using the MIMIC-III dataset. MIMIC-IV results will be added soon.

Table 4: Prediction Performance for Hospital Readmission and Mortality using MIMIC-III

| Model | Task 1: Mortality Prediction | | Task 2: Readmission Prediction | |
|---|---|---|---|---|
| | PRAUC (%) | ROCAUC (%) | PRAUC (%) | ROCAUC (%) |
| Transformer (Vaswani, 2017) | 10.17 | 56.91 | 66.22 | 62.75 |
| retain (Choi et al., 2016b) | 11.06 | 57.66 | 68.14 | 63.99 |
| GCT (Choi et al., 2020) | 10.48 | 58.99 | 68.16 | 65.48 |
| TCN (Bai et al., 2018) | 10.77 | 57.78 | 68.27 | 64.28 |
| GRASP (Zhang et al., 2021) | 10.75 | 58.72 | 69.70 | 65.24 |
| StageNet (Gao et al., 2020) | 10.62 | 57.79 | 68.20 | 65.22 |
| AdaCare Ma et al. (2020) | 11.00 | 58.77 | 68.73 | 64.86 |
| Deepr (Nguyen et al., 2016) | 11.18 | 59.74 | 69.63 | 65.59 |
| GRAM (Choi et al., 2017) | 12.27 | 58.50 | 68.32 | 64.36 |
| MMORE (Song et al., 2019) | 12.37 | 59.77 | 68.13 | 64.60 |
| KAME (Ma et al., 2018) | 12.15 | 57.98 | 67.89 | 63.69 |
| HAP (Zhang et al., 2020) | 11.15 | 57.10 | 67.82 | 63.85 |
| ARCI (Hadizadeh Moghaddam et al., 2024) | 11.93 | 60.19 | 68.03 | 65.17 |
| HyTransf (Xu et al., 2023) | 12.31 | 57.63 | 67.51 | 63.30 |
| **OntoFAR** | **14.39** | **63.69** | **70.41** | **66.15** |

Table 5: Performance comparison on Sequential Diagnosis Prediction based on PR-AUC, F1 score, and Acc@20.

| D | Model | General Performance | | | Label Category Performance (AUPRC) | | | |
|---|---|---|---|---|---|---|---|---|
| | | PR-AUC | F1 | Acc@20 | 0-25% | 25-50% | 50-75% | 75-100% |
| MIMIC-III | Transformer (Vaswani, 2017) | 28.23 | 22.36 | 38.03 | 28.07 | 54.20 | 50.62 | 74.14 |
| | GRAM (Choi et al., 2017) | 28.99 | 23.62 | 39.14 | 28.84 | 54.58 | 50.58 | 74.44 |
| | MMORE (Song et al., 2019) | 29.11 | 23.67 | 39.14 | 28.87 | 54.92 | 51.29 | 74.42 |
| | KAME (Ma et al., 2018) | 28.52 | 23.18 | 38.36 | 28.19 | 55.13 | 50.09 | 73.79 |
| | HAP (Zhang et al., 2020) | 29.28 | 23.25 | 39.46 | 29.10 | 55.11 | 52.19 | 76.28 |
| | HyTransformer (Xu et al., 2023) | 28.54 | 24.17 | 38.86 | 28.48 | 53.64 | 52.47 | 76.79 |
| | ARCI (Hadizadeh Moghaddam et al., 2024) | 29.19 | 25.84 | 39.06 | 28.88 | 53.95 | 52.49 | 76.61 |
| | Model (w/ GAT) | **30.43** | **26.25** | **40.80** | **30.18** | **56.23** | 52.93 | **76.97** |
| | Model (w/ HAT) | 30.27 | 26.05 | 40.52 | 30.08 | 55.67 | **53.22** | 76.64 |