# OpenReview forum: "OntoFAR: Hierarchical Multi-Ontology Fusion Better Augments EHR Representation"
_ICLR.cc/2025/Conference — Submitted to ICLR 2025_

### Official Review · Reviewer_8VUN · 2024-11-01

**Soundness:** 3
**Presentation:** 2
**Contribution:** 2
**Rating:** 5
**Confidence:** 3

**Summary:**

The paper introduces OntoFAR, an innovative framework designed to enhance the representation of medical concepts by integrating multiple medical ontology graphs. These graphs typically structure medical knowledge hierarchically and relate it to the medical codes used in electronic health records (EHRs). Current methods are limited by their inability to cross-reference information across different ontological and are restricted to using relationships within the same ontology. OntoFAR overcomes the limitations of previous approaches by fusing multiple ontologies. This is achieved through both vertical and horizontal message passing—vertical across different levels of the ontology hierarchy and horizontal across co-occurring concepts within EHR visits.

**Strengths:**

Originality: The OntoFAR framework introduces a novel approach to integrating multiple medical ontology graphs, enabling both horizontal and vertical message passing across these ontologies. This method of cross-ontology relationship exploitation is innovative in its bidirectional propagation mechanism, which is distinct from existing works that typically focus on single ontology systems or unidirectional information flow.

Quality: The paper is methodologically rigorous, presenting a clear and systematic approach to the multi-ontology integration problem. The proposed framework, OntoFAR, is well-constructed with detailed descriptions of its components.

Significance: The significance of this work lies in its potential to enhance the accuracy and robustness of predictive healthcare models by leveraging a richer representation of medical concepts derived from multiple ontologies.

**Weaknesses:**

1) The main weakness of the paper lies in its evaluation:

- The approach is evaluated solely on the task of sequential diagnosis prediction, which restricts the assessment of OntoFAR's effectiveness. To explore the benefits of integrating multiple medical ontology graphs for different types of medical concepts, the approach should be tested on a variety of tasks that utilize diverse concepts.

- The experiments are conducted on MIMIC-III and MIMIC-IV, which originate from the same healthcare provider (Beth Israel Deaconess Medical Center in Boston). Additionally, both databases contain overlapping admissions from 2008 to 2012, which limits the diversity of the data used in the study. Evaluating the approach in other datasets is essential for understanding its applicability and robustness across different settings.

- When comparing the performance of OntoFAR with other existing medical ontology structure encoders, authors should also indicate the size of each model (number of trainable weights). The observed performance increase for OntoFAR may be due to an enlargement of the model relative to other methods, rather than its effectiveness in learning superior medical concept representations.

2) In general, the writing requires significant improvements. For example, line 171: “work depicted Figure 1” should be corrected to “work depicted in Figure 1”; additionally, there are multiple typographical errors, such as the use of “massage” instead of “message” multiple times.

**Questions:**

- Authors need to clarify the meaning of “Pr” in equation 1.

- Additional clarification on how to aggregate clinical events embeddings from the matrix “Z” to obtain a single visit representation is needed (lines 333-335).

- In line 442, the authors likely mean “Table 3” instead of “Figure 3”. Similarly, in line 500, they might intend to refer to “Right” instead of “Left”.

- The authors should describe the elements in Fig. 6, such as the meaning of the nodes based on their color, to enhance understanding of the figure’s components.

---

> ### Author Response · Authors · 2024-11-25
>
> Dear Reviewer 8VUN,
>
> Thank you for your thoughtful review and valuable feedback. Below, we address your questions and provide new experimental results to incorporate your recommendations and address your concerns.
>
>
>
> 1.  **The approach is evaluated solely on the task of sequential diagnosis prediction**
>
> We added two more clinical tasks: 1) mortality prediction and 2) hospital readmission prediction in additional to diagnosis prediction. Table A2 presents the results of these tasks alongside baseline comparisons, based on the MIMIC-III dataset. Experiments on the MIMIC-IV dataset are underway, and we will share the results once completed.
>
> In addition, we also added more relevant baselines for mortality prediction and readmission prediction:
>
> [1] Nguyen, Phuoc, et al. "Deepr: a convolutional net for medical records (2016)." ArXiv160707519 Cs Stat (2016).
>
> [2] Bai, Shaojie, J. Zico Kolter, and Vladlen Koltun. "An empirical evaluation of generic convolutional and recurrent networks for sequence modeling." arXiv preprint arXiv:1803.01271 (2018).
>
> [3] Song, Lihong, et al. "Medical Concept Embedding with Multiple Ontological Representations." IJCAI. Vol. 19. 2019.
>
> [4] Choi, Edward, et al. "Learning the graphical structure of electronic health records with graph convolutional transformer." Proceedings of the AAAI conference on artificial intelligence. Vol. 34. No. 01. 2020.
>
> [5] Gao, Junyi, et al. "Stagenet: Stage-aware neural networks for health risk prediction." Proceedings of The Web Conference 2020. 2020.
>
> [6] Ma, Liantao, et al. "Adacare: Explainable clinical health status representation learning via scale-adaptive feature extraction and recalibration." Proceedings of the AAAI Conference on Artificial Intelligence. Vol. 34. No. 01. 2020.
>
> [7] Zhang, Chaohe, et al. "GRASP: generic framework for health status representation learning based on incorporating knowledge from similar patients." Proceedings of the AAAI conference on artificial intelligence. Vol. 35. No. 1. 2021.
>
> [8] Xu, Ran, et al. "Hypergraph transformers for ehr-based clinical predictions." AMIA Summits on Translational Science Proceedings 2023 (2023): 582.
>
> [9] Hadizadeh Moghaddam, Arya, et al. "Contrastive Learning on Medical Intents for Sequential Prescription Recommendation." Proceedings of the 33rd ACM International Conference on Information and Knowledge Management. 2024.
>
>
> ### Table A2: Prediction Performance for Hospital Readmission and Mortality
> | Model           | **PRAUC (Readmission)** | **ROCAUC (Readmission)** | **PRAUC (Mortality)** | **ROCAUC (Mortality)** |
> |-----------------|-----------------------------|------------------------------|---------------------------|----------------------------|
> | Transformer     | 66.22                      | 62.75                       | 10.17                    | 56.91                     |
> | retain          | 68.14                      | 63.99                       | 11.06                    | 57.66                     |
> | GCT [4]         | 68.16                      | 65.48                       | 10.48                    | 58.99                     |
> | TCN [2]         | 68.27                      | 64.28                       | 10.77                    | 57.78                     |
> | GRASP [7]       | 69.70                      | 65.24                       | 10.75                    | 58.72                     |
> | StageNet [5]    | 68.20                      | 65.22                       | 10.62                    | 57.79                     |
> | AdaCare [6]     | 68.73                      | 64.86                       | 11.00                    | 58.77                     |
> | Deepr [1]       | 69.63                      | 65.59                       | 11.18                    | 59.74                     |
> | GRAM            | 68.32                      | 64.36                       | 12.27                    | 58.50                     |
> | MMORE           | 68.13                      | 64.60                       | 12.37                    | 59.77                     |
> | KAME            | 67.89                      | 63.69                       | 12.15                    | 57.98                     |
> | HAP             | 67.82                      | 63.85                       | 11.15                    | 57.10                     |
> | ARCI [9]        | 68.03                      | 65.17                       | 11.93                    | 60.19                     |
> | HyTransformer [8]| 67.51                     | 63.30                       | 12.31                    | 57.63                     |
> | **OntoFAR**     | **70.41**                  | **66.15**                   | **14.39**                | **63.69**                 |

---

> ### Author Response · Authors · 2024-11-25
>
> 2.  **The experiments are conducted on MIMIC-III and MIMIC-IV, which originate from the same healthcare provider. More datasets needed.**
>
> While MIMIC-III and MIMIC-IV share some patient overlap, they differ significantly in critical aspects, making their joint use valuable. MIMIC-III covers ICU stays from 2001 to 2012, while MIMIC-IV spans 2008 to 2019 and includes newer coding systems like ICD-10-CM/PCS. MIMIC-IV also adopts a modular structure with enhanced metadata. We also considered including the eICU Collaborative Research Database. However, many features, such as medications, are recorded using generic names rather than standardized systems like RxNorm or ATC. Since our study focuses on integrating different ontologies for diagnoses, drugs, and procedures, we decided not to include eICU, as it lacks an ontology system for each of these features.
>
>
>
>
> 3.  **The experiments are conducted on MIMIC-III and MIMIC-IV, which originate from the same healthcare provider. More datasets needed.**
>
> While MIMIC-III and MIMIC-IV share some patient overlap, they differ significantly in critical aspects, making their joint use valuable. MIMIC-III covers ICU stays from 2001 to 2012, while MIMIC-IV spans 2008 to 2019 and includes newer coding systems like ICD-10-CM/PCS. MIMIC-IV also adopts a modular structure with enhanced metadata. We also considered including the eICU Collaborative Research Database. However, many features, such as medications, are recorded using generic names rather than standardized systems like RxNorm or ATC. Since our study focuses on integrating different ontologies for diagnoses, drugs, and procedures, we decided not to include eICU, as it lacks an ontology system for each of these features.
>
>
>
> 4.  **In general, the writing requires significant improvements. For example, line 171: “work depicted Figure 1” should be corrected to “work depicted in Figure 1”; additionally, there are multiple typographical errors, such as the use of “massage” instead of “message” multiple times.**
>
> Thanks for pointing this our. We fix it in the paper.
>
>
>
>
> 5.  **Authors need to clarify the meaning of “Pr” in equation 1.**
>
> Thanks for pointing that our. By ‘Pr’ we simply mean the prompt. So Pr(c) denotes the generated prompt for code c. We will clarify this in the paper.

---

> ### Author Response · Authors · 2024-11-25
>
> 6.  **Additional clarification on how to aggregate clinical events embeddings from the matrix “Z” to obtain a single visit representation is needed (lines 333-335).**
>
> For each visit $V_{t}$, we obtain a representation $\mathbf{v}_t \in \mathbb{R}^{d}$ by multiplying the final embedding matrix $\mathbf{Z}$ with a multi-hot vector $\mathbf{u}_t = \{0,1\}^{N^{(L)}}$, which represents clinical events existence in the visit. This operation simply sums up the representation vectors of all the codes inside a visit to get to the visit embedding.
>
> 7. **In line 442, the authors likely mean “Table 3” instead of “Figure 3”. Similarly, in line 500, they might intend to refer to “Right” instead of “Left”.**
>
> Thanks for pointing this our. We fix it in the paper.
>
> **1.**  **The authors should describe the elements in Fig. 6, such as the meaning of the nodes based on their color, to enhance understanding of the figure’s components.**
>
> Figure 6 presents an interpretative case study for OntoFAR, illustrating how it learns the representation of ICD-9 diagnosis code 428.0, which corresponds to _"Congestive heart failure, unspecified" (CHF)_. The following points provide clarification:
>
> -   The figure depicts the extracted sub-knowledge graph (sub-KG) for ICD-9 code 428.0 within the Meta-KG framework.
> -   Each hierarchical level contains diverse medical codes, represented as follows: red circles for procedures, green circles for diagnoses, and blue circles for drugs.
> -   Larger circles indicate higher-level codes, representing more general concepts compared to lower-level codes.
> -   Within each hierarchical level, solid blue lines connect the codes, representing co-occurrence-driven edges for horizontal message passing.
> -   Dashed lines illustrate parent-child relationships for bottom-up vertical message passing (HGIP), with color coding (red for procedures, green for diagnoses, and blue for drugs). Parent-child edges for top-down GRAM propagation are depicted using solid black lines.
> -   The edge weights during horizontal and vertical propagation are learned through attention mechanisms. The thickness of the edges in the figure reflects the relative attention assigned to each edge.

---

> ### Author Response · Authors · 2024-12-03
> **Additional Experimentation for New tasks of Mortality Prediction and Readmission Prediction with MIMIC-IV dataset.**
>
> Dear Reviewer 8VUN,
>
> Thank you for your detailed and thoughtful initial review. We have carefully addressed your concerns in our rebuttal. If possible, could you kindly review our response and share any additional thoughts or feedback you may have?
>
> Also, as we promised in our earlier response, we also completed the MIMIC-IV experimentation for the additional tasks of mortality prediction and readmission prediction (results are percentage %):
>
> | Model         | PRAUC (Readmission) | ROCAUC (Readmission) | PRAUC (Mortality) | ROCAUC (Mortality) |
> |---------------|---------------------|----------------------|-------------------|--------------------|
> | Transformer   | 63.08              | 62.45               | 3.05             | 60.22             |
> | retain        | 64.72              | 63.48               | 4.95             | 60.32             |
> | GCT           | 64.11              | 63.05               | 3.78             | 60.3              |
> | TCN           | 64.49              | 63.31               | 3.9              | 60.62             |
> | GRASP         | 66.08              | 64.78               | 4.21             | 57.44             |
> | StageNet      | 65.09              | 64.49               | 4.38             | 64.99             |
> | AdaCare       | 64.88              | 63.83               | 4.7              | 63.38             |
> | Deepr         | 67.23              | 65.01               | 3.7              | 64.79             |
> | GRAM          | 64.77              | 63.98               | 3.3              | 64.15             |
> | MMORE         | 65.02              | 63.87               | 4.01             | 62.24             |
> | KAME          | 64.11              | 63.31               | 3.2              | 61.11             |
> | HAP           | 65.4               | 64.1                | 3.83             | 63.03             |
> | ARCI          | 65.95              | 64.33               | 3.97             | 63.41             |
> | HyTransformer | 65.55              | 64.5                | 4.58             | 58.43             |
> | **OntoFAR**   | **68.17**          | **65.11**           | **8.23**         | **66.35**         |
>
>
> We sincerely appreciate your thoughtful feedback and have taken great care to thoroughly address your concerns. If you have any further comments or suggestions, we would be delighted to consider them to further improve the quality of our submission. Thank you once again for your time and invaluable insights!

---

### Official Review · Reviewer_xUeg · 2024-11-02

**Soundness:** 3
**Presentation:** 3
**Contribution:** 3
**Rating:** 5
**Confidence:** 4

**Summary:**

The paper proposes a new unified framework that incorporates hierarchical information from multiple ontologies to augment patient representation in electronic health records. The idea is by incorporating multiple ontologies, the model can leverage cross-ontology relationships to fully leverage existing medical knowledge bases. The framework is evaluated against various baselines on two different datasets.

**Strengths:**

* A unified framework for incorporating multiple ontologies where each ontology can have different hierarchical structures.
* Extensive experiments using different graph backbones (GAT and HAT), different diagnosis prediction models (transformer, RETAIN, and TCN), and 2 datasets (although there is some overlap between the two datasets).
* A case study to demonstrate how the hierarchical and co-occurrence codes can help learn a better embedding representation.

**Weaknesses:**

* The paper fails to mention and benchmark against ADORE (Adaptive Integration of Categorical and Multi-relationalOntologies with EHR Data for Medical Concept Embedding by Cheong et al. 2023) which incorporates a multi-relational medical ontology, SNOMED-CT which combines medications and diagnoses into a single representation.
* There are different knowledge bases such as SNOMED-CT, CCS, and several others even for the diagnosis. Is there a reason why only one knowledge base is explored for diagnosis and/or medication (SNOMED-CT works for medication as well)?
* Only a single downstream task is benchmarked and OntoFAR is introduced as being beneficial for a variety of tasks. How does the embedding perform on other tasks like mortality prediction or readmission prediction for either of the datasets (it does not need to be both)?
* Given MIMIC-III and MIMIC-IV share the same dataset, it would be helpful to benchmark against something that is likely to have different patients. eICU is a good example of a potential open-source dataset (there is some shared with MIMIC but there is also some outside ones).
* The methodology section is quite dense and a bit hard to parse especially when trying to ascertain how information is shared across the different ontologies. It would be helpful to provide an example using ATC and ICD-9 hierarchy. Based on Figure 1, GAT or HAT are the mechanisms for sharing information across the same concept level across ontologies but this isn't made explicit.
* The citation style is incorrect, you should swap it to \citep as the default instead of \cite.

**Questions:**

* Why are OpenAI off-the-shelf LLMs used when there are many other open-source LLMs available? How sensitive is the performance to the quality of the embeddings from the LLM?
* How does your model compare against ADORE?

---

> ### Author Response · Authors · 2024-11-25
>
> Dear Reviewer xUeg,
>
> Thank you for your thoughtful review and valuable feedback. Below, we address your questions and provide new experimental results to incorporate your recommendations and address your concerns.
>
>
> 1.  **Only a single downstream task is benchmarked.**
>
> We added two more clinical tasks: 1) mortality prediction and 2) hospital readmission prediction in additional to diagnosis prediction. Table A2 presents the results of these tasks alongside baseline comparisons, based on the MIMIC-III dataset. Experiments on the MIMIC-IV dataset are underway, and we will share the results once completed.
>
> In addition, we also added more relevant baselines for mortality prediction and readmission prediction:
>
> [1] Nguyen, Phuoc, et al. "Deepr: a convolutional net for medical records (2016)." ArXiv160707519 Cs Stat (2016).
>
> [2] Bai, Shaojie, J. Zico Kolter, and Vladlen Koltun. "An empirical evaluation of generic convolutional and recurrent networks for sequence modeling." arXiv preprint arXiv:1803.01271 (2018).
>
> [3] Song, Lihong, et al. "Medical Concept Embedding with Multiple Ontological Representations." IJCAI. Vol. 19. 2019.
>
> [4] Choi, Edward, et al. "Learning the graphical structure of electronic health records with graph convolutional transformer." Proceedings of the AAAI conference on artificial intelligence. Vol. 34. No. 01. 2020.
>
> [5] Gao, Junyi, et al. "Stagenet: Stage-aware neural networks for health risk prediction." Proceedings of The Web Conference 2020. 2020.
>
> [6] Ma, Liantao, et al. "Adacare: Explainable clinical health status representation learning via scale-adaptive feature extraction and recalibration." Proceedings of the AAAI Conference on Artificial Intelligence. Vol. 34. No. 01. 2020.
>
> [7] Zhang, Chaohe, et al. "GRASP: generic framework for health status representation learning based on incorporating knowledge from similar patients." Proceedings of the AAAI conference on artificial intelligence. Vol. 35. No. 1. 2021.
>
> [8] Xu, Ran, et al. "Hypergraph transformers for ehr-based clinical predictions." AMIA Summits on Translational Science Proceedings 2023 (2023): 582.
>
> [9] Hadizadeh Moghaddam, Arya, et al. "Contrastive Learning on Medical Intents for Sequential Prescription Recommendation." Proceedings of the 33rd ACM International Conference on Information and Knowledge Management. 2024.
>
>
> ### Table A2: Prediction Performance for Hospital Readmission and Mortality
> | Model           | **PRAUC (Readmission)** | **ROCAUC (Readmission)** | **PRAUC (Mortality)** | **ROCAUC (Mortality)** |
> |-----------------|-----------------------------|------------------------------|---------------------------|----------------------------|
> | Transformer     | 66.22                      | 62.75                       | 10.17                    | 56.91                     |
> | retain          | 68.14                      | 63.99                       | 11.06                    | 57.66                     |
> | GCT [4]         | 68.16                      | 65.48                       | 10.48                    | 58.99                     |
> | TCN [2]         | 68.27                      | 64.28                       | 10.77                    | 57.78                     |
> | GRASP [7]       | 69.70                      | 65.24                       | 10.75                    | 58.72                     |
> | StageNet [5]    | 68.20                      | 65.22                       | 10.62                    | 57.79                     |
> | AdaCare [6]     | 68.73                      | 64.86                       | 11.00                    | 58.77                     |
> | Deepr [1]       | 69.63                      | 65.59                       | 11.18                    | 59.74                     |
> | GRAM            | 68.32                      | 64.36                       | 12.27                    | 58.50                     |
> | MMORE           | 68.13                      | 64.60                       | 12.37                    | 59.77                     |
> | KAME            | 67.89                      | 63.69                       | 12.15                    | 57.98                     |
> | HAP             | 67.82                      | 63.85                       | 11.15                    | 57.10                     |
> | ARCI [9]        | 68.03                      | 65.17                       | 11.93                    | 60.19                     |
> | HyTransformer [8]| 67.51                     | 63.30                       | 12.31                    | 57.63                     |
> | **OntoFAR**     | **70.41**                  | **66.15**                   | **14.39**                | **63.69**                 |

---

> ### Author Response · Authors · 2024-11-25
>
> **2.**  **There are different knowledge bases such as SNOMED-CT, CCS, and several others even for the diagnosis. Is there a reason why only one knowledge base is explored for diagnosis and/or medication (SNOMED-CT works for medication as well)?**
>
> Our model can be adopted on many ontology structures including, ICD9, ICD10, SNOMET, and CCS. We use ICD for diagnosis because it is currently the most widely used ontology in EHR datasets including MIMIC. Besides, ICD provides the rich hierarchies we need to fuse with other ontologies systems.
>
>
> **3.** **Given MIMIC-III and MIMIC-IV share the same dataset, it would be helpful to benchmark against something that is likely to have different patients. eICU is a good example of a potential open-source dataset.**
>
> While MIMIC-III and MIMIC-IV may share some patient overlap (we can’t verify it since all patients are deidentified), they differ significantly in critical aspects. MIMIC-III covers ICU stays from 2001 to 2012 (46,000 patients), while MIMIC-IV spans 2008 to 2019 (383,220 patients) and adopts a newer coding system like ICD-10-CM/PCS.
>
> Although eICU Collaborative Research Database is a another widely used public EHR dataset, the key features of medications in eICU are recorded using generic names rather than standardized ontology concepts like RxNorm or ATC. Since our study focuses on integrating different ontologies for diagnoses, drugs, and procedures, eICU lacks an ontology system for each of these features.
>
> **4.** **The methodology section is quite dense and a bit hard to parse especially when trying to ascertain how information is shared across the different ontologies. It would be helpful to provide an example using ATC and ICD-9 hierarchy. Based on Figure 1, GAT or HAT are the mechanisms for sharing information across the same concept level across ontologies but this isn't made explicit.**
>
> We included an algorithm in appendix A.1 which demonstrates the step-by-step process of medical code representation learning in OntoFAR. Also, Figure 6 illustrates an example of how OntoFAR learns the representation of the ICD-9 code 428.0, which denotes "Congestive heart failure, unspecified" (CHF).
>
> **5.** **The citation style is incorrect, you should swap it to \citep as the default instead of \cite.**
>
> Thank you very much for pointing out that out. We fixed this error.
>
> **6.** **Why are OpenAI off-the-shelf LLMs used when there are many other open-source LLMs available? How sensitive is the performance to the quality of the embeddings from the LLM?**
>
> OpenAI’s LLMs, such as GPT, are renowned for their state-of-the-art performance and extensive pretraining on diverse datasets. Also, they provide a standardized API to retrieve embeddings. This ensures high-quality and efficient general knowledge embeddings, which are particularly valuable when initializing domain-specific representations.

---

> ### Author Response · Authors · 2024-11-25
>
> **7.** **How does your model compare against ADORE, which incorporates a multi-relational medical ontology, SNOMED-CT which combines medications and diagnoses into a single representation.**
>
> Our work and ADORE have key differences in method:
>
> - In horizontal propagation, ADORE relies on the SNOMED ontology to capture interactions between diagnosis and drug codes (excluding procedure codes) using existed relational edges. However, this approach is based on an ontology structure (external source) and does not fully reflect interactions among medical codes from different sources as observed in EHRs reality. In contrast, OntoFAR captures these interactions more effectively by:
>
> - - Connecting codes from different sources at all levels of hierarchical ontologies, using co-occurrence information inherent in the dataset to reflect EHR realities. (Connecting these multi-source ontologies using EHRs reality, which lead to Mining EHR patterns holistically, integrating them with ontology structure learning.) in other words, we captures these multisource interaction by taking into account a synergy of EHRs (real statistical information) and Ontology (predefined medical knowledge).
>
> - - Deriving associations across varying granularities, integrating fine-grained and coarse-grained information for enhanced representation learning.
>
> - In vertical code encoding, ADORE relies solely on the top-down propagation of GRAM, which does not account for the hierarchy's order. It combines parent codes with child nodes using attention to generate embeddings but does not distinguish between parents or incorporate information about children into parent embeddings. In contrast, OntoFAR introduces the Hierarchical Graph Information Propagation (HGIP) module. This module propagates information upwards through a bottom-up sequence of attention subgraphs across consecutive ontology levels. The final code representation is then generated using a GRAM module, ensuring a more comprehensive encoding of hierarchical relationships.
>
> We were unable to implement the GitHub code for this baseline within the limited time available to include it in our experiments. However, we will acknowledge this valuable work in our related work section and include a discussion about it. To address this issue, we added 2 more recent baselines:
>
> [8] Xu, Ran, et al. "Hypergraph transformers for ehr-based clinical predictions." AMIA Summits on Translational Science Proceedings 2023 (**2023**): 582.
>
> [9] Hadizadeh Moghaddam, Arya, et al. "Contrastive Learning on Medical Intents for Sequential Prescription Recommendation." Proceedings of the 33rd ACM International Conference on Information and Knowledge Management. **2024**.
>
> Please refer to Table A1 below.
>
> # Table A1: Sequential Diagnosis Prediction
> | Model                      | PR-AUC  | F1     | Acc@20  | 0-25%   | 25-50%  | 50-75%  | 75-100% |
> |----------------------------|---------|--------|---------|---------|---------|---------|---------|
> | Transformer                | 28.23   | 22.36  | 38.03   | 28.07   | 54.20   | 50.62   | 74.14   |
> | GRAM                       | 28.99   | 23.62  | 39.14   | 28.84   | 54.58   | 50.58   | 74.44   |
> | MMORE                      | 29.11   | 23.67  | 39.14   | 28.87   | 54.92   | 51.29   | 74.42   |
> | KAME                       | 28.52   | 23.18  | 38.36   | 28.19   | 55.13   | 50.09   | 73.79   |
> | HAP                        | 29.28   | 23.25  | 39.46   | 29.10   | 55.11   | 52.19   | 76.28   |
> | HyTransformer              | 28.54   | 24.17  | 38.86   | 28.48 | 53.64  | 52.47   | 76.79   |
> | ARCI                       |  29.19   | 25.84  | 39.06   | 28.88 | 53.95  | 52.49   | 76.61   |
> | Model (w/ GAT)             | **30.43** | **26.25** | **40.80** | **30.18** | **56.23** | 52.93   | **76.97** |
> | Model (w/ HAT)             | 30.27   | 26.05  | 40.52   | 30.08   | 55.67   | **53.22** | 76.64   |

---

> > ### Comment · Reviewer_xUeg · 2024-11-26
> > **Response to Initial Author's response**
> >
> > Thank you for the detailed response. I'm waiting for the updated paper to better understand some of the changes and to assess the work again. I do want to highlight a few aspects of the responses that still need additional clarification:
> >
> > * The comment is that ICD is the most common ontology in EHR systems is somewhat debatable. While I agree that ICD is the default vocabulary, there are many existing works that have demonstrated ICD often has poor precision and recall. As such, many other ontologies (SNOMED, CCS, DRG) have all been adopted to group ICD codes in different manners. As an example, CCS will bridge across different ICD categories for septicemia (see https://hcup-us.ahrq.gov/toolssoftware/ccs/AppendixASingleDX.txt). As such, providing additional experimental validation across different ontologies can help demonstrate the work has additional benefits that is not single-ontology specific.
> > * Regarding the use of ChatGPT, there has been substantial work in the medical LLM space (e.g., Meditron, MedLlama, etc.) that have fine-tuned the model to work better for medical domain. Is the proposed work only seeing the benefits of incorporating ontology because ChatGPT hasn't been trained on such data? If so, is incorporating the ontology sufficient to outperform medical domain specific LLMs?
> > * I appreciate the specifics of how the work can be distinguished from ADORE but its's still not entirely clear whether these key differences can provide performance benefits that outweigh the additional computation associated with integrating multi-ontologies. Having some empirical results using the SNOMED ontology might help allay this concern.

---

> > > ### Author Response · Authors · 2024-11-28
> > >
> > > **1. The comment is that ICD is the most common ontology in EHR systems is somewhat debatable. While I agree that ICD is the default vocabulary, there are many existing works that have demonstrated ICD often has poor precision and recall. As such, many other ontologies (SNOMED, CCS, DRG) have all been adopted to group ICD codes in different manners. As an example, CCS will bridge across different ICD categories for septicemia. As such, providing additional experimental validation across different ontologies can help demonstrate the work has additional benefits that is not single-ontology specific.**
> > >
> > > Dear reviewer xUeg, thank you for the suggestion on alternative ontology systems for our work. Given the ICD vocabulary as the default ontology in major public EHR datasets such as MIMIC III & IV, eICU, our aim is not to identify a superior ontology for these datasets but rather to address a pressing challenge of multi-ontology integration: when existing EHR datasets are associated with multiple ontologies, how can we effectively fuse them in a comprehensive manner to facilitate ontology-guided embedding?
> > > We acknowledge the suggestion of using the CCS ontology. However, CCS has limitations compared to ICD for our purposes. Specifically, CCS has a relatively shallow and flat structure, with mostly a single level of concept classification and many small groups. While our model is adaptable to such structures, it cannot fully leverage the rich hierarchical relationships and categorizations present in ICD. Therefore, it is not motivated to replace the default ICD with CCS in our work.
> > > Regarding the concern about single-ontology specification, we wanted to emphasize that our model is inherently designed to adopt and fuse multiple ontologies. It has been validated on such use cases. We regret that we could not implement additional experiments within the rebuttal timeframe, such as incorporating SNOMED-based baselines (e.g., ADORE) as you suggested. These experiments remain an important direction for future work.
> > >
> > > **2. Regarding the use of ChatGPT, there has been substantial work in the medical LLM space (e.g., Meditron, MedLlama, etc.) that have fine-tuned the model to work better for medical domain. Is the proposed work only seeing the benefits of incorporating ontology because ChatGPT hasn't been trained on such data? If so, is incorporating the ontology sufficient to outperform medical domain specific LLMs?**
> > >
> > > We chose ChatGPT over medical specific LLM because our model doesn't rely on LLM for specific prediction. Instead, we utilize LLM solely to generate contextually enriched initial embeddings of ontology concepts, which are subsequently refined through our ontology fusion and learning framework. From this perspective, the specific choice of LLM is not central to the contribution of our work. The focus of this study lies in the integration of multiple ontology systems applied to the EHR embeddings, rather than the characteristics of using differently LLMs to generate them.
> > >
> > > **3. I appreciate the specifics of how the work can be distinguished from ADORE but its's still not entirely clear whether these key differences can provide performance benefits that outweigh the additional computation associated with integrating multi-ontologies. Having some empirical results using the SNOMED ontology might help allay this concern.**
> > >
> > > We appreciate the reviewer’s suggestion to include empirical result using SNOMED, however, we would like to address some challenges of implementing SNOMED in our work. MIMIC III and IV datasets primarily rely on ICD coding systems. As a result, using SNOMED system on these datasets will require translation (e.g., the ICD9 to SNOMED-CT mapping is also noted in ADORE Pre-processing section). While SNOMED provide concept mapping instruction, the translation process introduces complexities. Around 30% of ICD9 codes involve one-to-many mappings, and around 6% have no direct mapping to SNOMED concepts. This results in a significant portion of the codes being either ambiguously translated or completely untranslatable. The validity of the information in original EHR dataset are potentially compromised.
> > > While we made efforts to include ADORE as a baseline in our experiments, we were unable to finalize its implementation within the available timeframe. However, we acknowledge the importance of ADORE and will emphasize its significance in the related work.

---

> ### Author Response · Authors · 2024-12-03
> **Additional Experimentation for New tasks of Mortality Prediction and Readmission Prediction with MIMIC-IV dataset.**
>
> Dear Reviewer xUeg,
>
> Thank you for your detailed and thoughtful initial review. We have carefully addressed your concerns in our rebuttal. If convenient, could you kindly review our response and share any additional thoughts or feedback? Your input would be greatly appreciated!
>
> Also, as we promised in our earlier response, we also completed the MIMIC-IV experimentation for the additional tasks of mortality prediction and readmission prediction (results are percentage %):
>
> | Model         | PRAUC (Readmission) | ROCAUC (Readmission) | PRAUC (Mortality) | ROCAUC (Mortality) |
> |---------------|---------------------|----------------------|-------------------|--------------------|
> | Transformer   | 63.08              | 62.45               | 3.05             | 60.22             |
> | retain        | 64.72              | 63.48               | 4.95             | 60.32             |
> | GCT           | 64.11              | 63.05               | 3.78             | 60.3              |
> | TCN           | 64.49              | 63.31               | 3.9              | 60.62             |
> | GRASP         | 66.08              | 64.78               | 4.21             | 57.44             |
> | StageNet      | 65.09              | 64.49               | 4.38             | 64.99             |
> | AdaCare       | 64.88              | 63.83               | 4.7              | 63.38             |
> | Deepr         | 67.23              | 65.01               | 3.7              | 64.79             |
> | GRAM          | 64.77              | 63.98               | 3.3              | 64.15             |
> | MMORE         | 65.02              | 63.87               | 4.01             | 62.24             |
> | KAME          | 64.11              | 63.31               | 3.2              | 61.11             |
> | HAP           | 65.4               | 64.1                | 3.83             | 63.03             |
> | ARCI          | 65.95              | 64.33               | 3.97             | 63.41             |
> | HyTransformer | 65.55              | 64.5                | 4.58             | 58.43             |
> | **OntoFAR**   | **68.17**          | **65.11**           | **8.23**         | **66.35**         |
>
>
> We are truly grateful for your thoughtful feedback and have worked diligently to address your concerns. If you have any further comments or suggestions, we would gladly consider them to refine and improve our submission. Thank you again for your time and invaluable insights!

---

### Official Review · Reviewer_HcPA · 2024-11-04

**Soundness:** 3
**Presentation:** 2
**Contribution:** 2
**Rating:** 6
**Confidence:** 4

**Summary:**

The authors propose a framework named OntoFAR for enhancing medical concept representation in EHRs by fusing multiple medical ontologies. It enables message passing both vertically and horizontally to capture richer cross-ontology relationships. OntoFAR constructs a unified Meta-KG initialized with embeddings from pre-trained language models, and effectively integrates medical concept information across ontologies. Evaluations on MIMIC-III and MIMIC-IV datasets demonstrate OntoFAR’s superior predictive performance and robustness, especially in data-limited scenarios, over existing EHR representation methods.

**Strengths:**

1. The paper addresses the crucial real-world issue of diagnosis prediction. They effectively integrate multiple ontologies to enhance predictions and resolve alignment challenges among different ontologies.
2. The authors present comprehensive experimental results from various perspectives, including many ablation studies and additional analyses that deepen the understanding of their findings.
3. The figures and tables in the paper are well-designed and contribute significantly to clarifying the framework and interpreting the results.

**Weaknesses:**

1. The comparison with the baselines does not seem to be fair. The proposed framework OntoFAR utilizes the GPT text embedding model for the embeddings of medical concepts, which is a much more powerful model than those used by the baselines. The ablation studies presented in Table 3 show that removing this part (w/o LLMs) results in performance on par with the baselines. This somehow suggests that OntoFAR might not truly outperform the baselines without the advantage of using this more powerful embedding model for a fair comparison.
2. The baselines used in the paper are ranging from 2017 to 2020, which are kind of outdated. It's better for the authors to consider more recent baselines, such as SeqCare [1] and other studies mentioned in the related works section (e.g., GraphCare, MedPath, RAM-EHR).
3. The notations in the method section are overly complex and could be much simplified. Many of the notations currently used are not essential.

Typos and formats:
- The referencing style throughout the paper is not correct; it should use parentheses (i.e., \citep{} instead of \cite{}).
- "Figure 3" in line 442 should be "Table 3"

[1] Xu, Yongxin, et al. "Seqcare: Sequential training with external medical knowledge graph for diagnosis prediction in healthcare data." Proceedings of the ACM Web Conference 2023. 2023.

**Questions:**

1. What's the base model in Figure 3?

---

> ### Author Response · Authors · 2024-11-24
>
> Dear Reviewer HcPA,
>
> Thank you for your thoughtful review and valuable feedback. Below, we address your questions and provide new experimental results to incorporate your recommendations and address your concerns.
>
>
> **1.**  **The comparison with the baselines does not seem to be fair. The proposed framework OntoFAR utilizes the GPT text embedding model for the embeddings of medical concepts, which is a much more powerful model than those used by the baselines.**
>
> Based on Table 3 in the paper (ablation study), This LLM embedding initialization improves the overall performance. However, without LLM initialization, our model still outperforms other baselines.
>
>
>
> **2.** **The referencing style throughout the paper is not correct; it should use parentheses (i.e., \citep{} instead of \cite{}). "Figure 3" in line 442 should be "Table 3"**
>
> Thank you for pointing these two issues out. We fix this issue and upload a new version of the manuscript.
>
> **3.**  **What's the base model in Figure 3?**
>
> In Figure 3, we aim to demonstrate the compatibility of OntoFAR as an advanced medical concept encoder. OntoFAR can be seamlessly integrated as an add-on to popular existing health predictive models, which can significantly enhance their performance. For example, Figure 3 presents the results of three predictive models (**base model**)—Transformer, TCN, and RETAIN—before and after incorporating OntoFAR, highlighting the performance improvements achieved with its integration.

---

> ### Author Response · Authors · 2024-11-24
>
> **4.**   **The baselines used in the paper range from 2017 to 2020, which are kind of outdated. It's better for the authors to consider more recent baselines, such as SeqCare [1] GraphCare, MedPath, RAM-EHR.**
>
> We acknowledge that RAM-EHR requires multi-source clinical text data, which makes it incompatible with the scope of our model that exclusively uses discrete EHR codes. For GraphCare, we implemented it using the code provided in their repository. However, the results were suboptimal due to limited instructions on running the code, an issue also noted in the GitHub repository's "Issues" section.
>
> Regarding SeqCare and MedPath, we attempted to implement them within the timeframe of the rebuttal period. Unfortunately, we could not get their GitHub code to function as required, primarily due to missing necessary files.
>
> To address this question and provide additional comparisons, we have incorporated two recent baselines into the sequential diagnosis prediction task. We hope this enhances the comprehensiveness of our evaluation.
>
> [8] Xu, Ran, et al. "Hypergraph transformers for ehr-based clinical predictions." AMIA Summits on Translational Science Proceedings 2023 (2023): 582.
>
> [9] Hadizadeh Moghaddam, Arya, et al. "Contrastive Learning on Medical Intents for Sequential Prescription Recommendation." Proceedings of the 33rd ACM International Conference on Information and Knowledge Management. 2024.
>
> Please refer to Table A1 below.
>
> # Table A1: Sequential Diagnosis Prediction
> | Model                      | PR-AUC  | F1     | Acc@20  | 0-25%   | 25-50%  | 50-75%  | 75-100% |
> |----------------------------|---------|--------|---------|---------|---------|---------|---------|
> | Transformer                | 28.23   | 22.36  | 38.03   | 28.07   | 54.20   | 50.62   | 74.14   |
> | GRAM                       | 28.99   | 23.62  | 39.14   | 28.84   | 54.58   | 50.58   | 74.44   |
> | MMORE                      | 29.11   | 23.67  | 39.14   | 28.87   | 54.92   | 51.29   | 74.42   |
> | KAME                       | 28.52   | 23.18  | 38.36   | 28.19   | 55.13   | 50.09   | 73.79   |
> | HAP                        | 29.28   | 23.25  | 39.46   | 29.10   | 55.11   | 52.19   | 76.28   |
> | HyTransformer              | 28.54   | 24.17  | 38.86   | 28.48 | 53.64  | 52.47   | 76.79   |
> | ARCI                       |  29.19   | 25.84  | 39.06   | 28.88 | 53.95  | 52.49   | 76.61   |
> | Model (w/ GAT)             | **30.43** | **26.25** | **40.80** | **30.18** | **56.23** | 52.93   | **76.97** |
> | Model (w/ HAT)             | 30.27   | 26.05  | 40.52   | 30.08   | 55.67   | **53.22** | 76.64   |

---

> ### Author Response · Authors · 2024-11-24
>
> To make the experimental section more comprehensive and extensive, we experiment with two more tasks of mortality prediction and readmission prediction. For each task we used all the baselines used in sequential diagnosis tasks plus more relevant baselines. Table A2 presents the results of these tasks based on the MIMIC-III dataset. Experiments on the MIMIC-IV dataset are underway, and we will share the results once completed.
>
> [1] Nguyen, Phuoc, et al. "Deepr: a convolutional net for medical records (2016)." ArXiv160707519 Cs Stat (2016).
>
> [2] Bai, Shaojie, J. Zico Kolter, and Vladlen Koltun. "An empirical evaluation of generic convolutional and recurrent networks for sequence modeling." arXiv preprint arXiv:1803.01271 (2018).
>
> [3] Song, Lihong, et al. "Medical Concept Embedding with Multiple Ontological Representations." IJCAI. Vol. 19. 2019.
>
> [4] Choi, Edward, et al. "Learning the graphical structure of electronic health records with graph convolutional transformer." Proceedings of the AAAI conference on artificial intelligence. Vol. 34. No. 01. 2020.
>
> [5] Gao, Junyi, et al. "Stagenet: Stage-aware neural networks for health risk prediction." Proceedings of The Web Conference 2020. 2020.
>
> [6] Ma, Liantao, et al. "Adacare: Explainable clinical health status representation learning via scale-adaptive feature extraction and recalibration." Proceedings of the AAAI Conference on Artificial Intelligence. Vol. 34. No. 01. 2020.
>
> [7] Zhang, Chaohe, et al. "GRASP: generic framework for health status representation learning based on incorporating knowledge from similar patients." Proceedings of the AAAI conference on artificial intelligence. Vol. 35. No. 1. 2021.
>
> [8] Xu, Ran, et al. "Hypergraph transformers for ehr-based clinical predictions." AMIA Summits on Translational Science Proceedings 2023 (2023): 582.
>
> [9] Hadizadeh Moghaddam, Arya, et al. "Contrastive Learning on Medical Intents for Sequential Prescription Recommendation." Proceedings of the 33rd ACM International Conference on Information and Knowledge Management. 2024.
>
> ### Table A2: Prediction Performance for Hospital Readmission and Mortality
> | Model           | **PRAUC (Readmission)** | **ROCAUC (Readmission)** | **PRAUC (Mortality)** | **ROCAUC (Mortality)** |
> |-----------------|-----------------------------|------------------------------|---------------------------|----------------------------|
> | Transformer     | 66.22                      | 62.75                       | 10.17                    | 56.91                     |
> | retain          | 68.14                      | 63.99                       | 11.06                    | 57.66                     |
> | GCT [4]         | 68.16                      | 65.48                       | 10.48                    | 58.99                     |
> | TCN [2]         | 68.27                      | 64.28                       | 10.77                    | 57.78                     |
> | GRASP [7]       | 69.70                      | 65.24                       | 10.75                    | 58.72                     |
> | StageNet [5]    | 68.20                      | 65.22                       | 10.62                    | 57.79                     |
> | AdaCare [6]     | 68.73                      | 64.86                       | 11.00                    | 58.77                     |
> | Deepr [1]       | 69.63                      | 65.59                       | 11.18                    | 59.74                     |
> | GRAM            | 68.32                      | 64.36                       | 12.27                    | 58.50                     |
> | MMORE           | 68.13                      | 64.60                       | 12.37                    | 59.77                     |
> | KAME            | 67.89                      | 63.69                       | 12.15                    | 57.98                     |
> | HAP             | 67.82                      | 63.85                       | 11.15                    | 57.10                     |
> | ARCI [9]        | 68.03                      | 65.17                       | 11.93                    | 60.19                     |
> | HyTransformer [8]| 67.51                     | 63.30                       | 12.31                    | 57.63                     |
> | **OntoFAR**     | **70.41**                  | **66.15**                   | **14.39**                | **63.69**                 |

---

> ### Comment · Reviewer_HcPA · 2024-11-27
>
> Thank you for the authors' response. My main concern remains that OntoFAR may rely on powerful GPT embeddings, and its performance without LLM embeddings does not show a statistically significant improvement over the baselines. Nevertheless, I appreciate the authors' efforts in providing additional analyses, and I am willing to raise my score to 6.

---

> > ### Author Response · Authors · 2024-11-27
> >
> > Dear Reviewer HcPA,
> > Thank you for taking the time to update your evaluation of our submission. We appreciate your thoughtful feedback. If you have any additional comments or suggestions, we are more than happy to address them to ensure the quality of our submission.

---

### Official Review · Reviewer_VcGN · 2024-11-04

**Soundness:** 3
**Presentation:** 3
**Contribution:** 3
**Rating:** 6
**Confidence:** 2

**Summary:**

The paper proposes OntoFAR, a framework that enhances EHR predictive modeling by integrating multiple medical ontologies. It introduces dual-dimensional message passing (vertical and horizontal) to enrich medical concept representations and uses LLMs for embedding initialization. The approach is validated on MIMIC-III and MIMIC-IV datasets, demonstrating superior performance over baselines and robustness in data-limited scenarios.

**Strengths:**

1. **Novel Idea with Cross-Ontology Integration**:

The proposed new method is able to capture relationships between different medical code types, enhancing representation.

2. **Robust Embedding Initialization**:

It is interesting and effective to leverage LLMs for enhanced concept embedding with external knowledge.

3. **Data Insufficiency Resilience**:

The author also performs additional experiments to prove that the proposed method maintains strong performance even with limited data availability.

**Weaknesses:**

**Substantive Assessment of Weaknesses**

1. **Insufficient Justification of Improvements and Potential Gaps in Related Work**:

While the paper proposes a multi-ontology framework to enhance EHR predictions, which is a well-explored domain, the authors' claims regarding the uniqueness and superiority of their approach are not convincingly substantiated. Integrating knowledge graphs (KGs) to improve EHR prediction is an established area with significant recent advancements. While the paper notes some limitations in current methods, it does not ensure that these criticisms translate into performance gains over state-of-the-art approaches. Notably, the comparison set lacks recent, relevant KG-based EHR prediction methods such as KerPrint [1], KAMPNet [2], and MedPath [3].

In particular, KAMPNet [2] presents a multi-source and multi-level graph framework similar in concept to the proposed OntoFAR, suggesting an overlap that should be clarified. To strengthen the paper, I recommend including these contemporary works as baselines to provide a comprehensive comparison. Additionally, a detailed discussion explaining how OntoFAR differs from and advances beyond KAMPNet’s multi-level graph strategy would be essential to highlight its distinct contributions.

**References**:
1. Yang K, Xu Y, Zou P, et al. *KerPrint: local-global knowledge graph enhanced diagnosis prediction for retrospective and prospective interpretations*. Proceedings of the AAAI Conference on Artificial Intelligence, 2023, 37(4): 5357-5365.
2. An Y, Tang H, Jin B, et al. *KAMPNet: multi-source medical knowledge augmented medication prediction network with multi-level graph contrastive learning*. BMC Medical Informatics and Decision Making, 2023, 23(1): 243.
3. Ye M, Cui S, Wang Y, et al. *Medpath: Augmenting health risk prediction via medical knowledge paths*. Proceedings of the Web Conference 2021. 2021: 1397-1409.

**Questions:**

1. What is the difference between the proposed work and the KAMPNet [2]?
2. The last tested baseline is HAP (2020) and there are many following works during the last four years why they are not included.


**References**:
1. Yang K, Xu Y, Zou P, et al. *KerPrint: local-global knowledge graph enhanced diagnosis prediction for retrospective and prospective interpretations*. Proceedings of the AAAI Conference on Artificial Intelligence, 2023, 37(4): 5357-5365.
2. An Y, Tang H, Jin B, et al. *KAMPNet: multi-source medical knowledge augmented medication prediction network with multi-level graph contrastive learning*. BMC Medical Informatics and Decision Making, 2023, 23(1): 243.
3. Ye M, Cui S, Wang Y, et al. *Medpath: Augmenting health risk prediction via medical knowledge paths*. Proceedings of the Web Conference 2021. 2021: 1397-1409.

---

> ### Author Response · Authors · 2024-11-24
>
> Dear Reviewer VcGN,
>
> Thank you for your thoughtful review and valuable feedback. Below, we address your questions and provide new experimental results to incorporate your recommendations and address your concerns.
>
> 1.  **What is the difference between the proposed work and the KAMPNet?**
>
> Our work and KAMPNet have key differences in method:
>
> 1. In vertical (homogenous) code encoding, KAMPNet models the whole hierarchy structure using one graph, which is unable to account for the order of node calculation by hierarchies. In OntoFAR, we develop the Hierarchical Graph Information Propagation (HGIP) module, where a bottom-up sequence of attention subgraphs on each consecutive ontology levels propagate information upwards, and finally a GRAM module generates the final code representation. Although both models ensure that parent nodes distill information from their descendants, OntoFAR explicitly controls the order of node embedding updates across the main graph.
>
> 2. In horizontal heterogenous (multi-source) code encoding, KAMPNet only captures the interaction of muti-source codes (diagnosis, medication) in the leaves code level of ontologies (at only one level of granularity). However, OntoFAR connects multiple ontologies (diagnosis, medication, procedures) across all levels of hierarchies. This approach captures concept co-occurrence across varying granularities, effectively integrating both fine-grained and coarse-grained information for representation learning.
>
> We add this valuable work in our related work section of our paper.

---

> ### Author Response · Authors · 2024-11-24
>
> 2.  **The last tested baseline is HAP (2020) and there are many following works during the last four years they are not included.**
>
> We made every effort to implement the three baselines you proposed. However, we encountered the following challenges:
>
> KAMPNet: The GitHub repository only includes processed data, and the implementation code is unavailable.
>
> KerPrint and MedPath: Both required additional input files that were missing in repo, preventing us from running the provided code in a short time.
>
> To address this limitation to some extent, we have included two additional recent baselines for the sequential diagnosis prediction task. These additions aim to enhance the comprehensiveness of our validation.
>
> [8] Xu, Ran, et al. "Hypergraph transformers for ehr-based clinical predictions." AMIA Summits on Translational Science Proceedings 2023 (**2023**): 582.
>
> [9] Hadizadeh Moghaddam, Arya, et al. "Contrastive Learning on Medical Intents for Sequential Prescription Recommendation." Proceedings of the 33rd ACM International Conference on Information and Knowledge Management. **2024**.
>
> Please refer to Table A1 below.
>
>
> # Table A1: Sequential Diagnosis Prediction
> | Model                      | PR-AUC  | F1     | Acc@20  | 0-25%   | 25-50%  | 50-75%  | 75-100% |
> |----------------------------|---------|--------|---------|---------|---------|---------|---------|
> | Transformer                | 28.23   | 22.36  | 38.03   | 28.07   | 54.20   | 50.62   | 74.14   |
> | GRAM                       | 28.99   | 23.62  | 39.14   | 28.84   | 54.58   | 50.58   | 74.44   |
> | MMORE                      | 29.11   | 23.67  | 39.14   | 28.87   | 54.92   | 51.29   | 74.42   |
> | KAME                       | 28.52   | 23.18  | 38.36   | 28.19   | 55.13   | 50.09   | 73.79   |
> | HAP                        | 29.28   | 23.25  | 39.46   | 29.10   | 55.11   | 52.19   | 76.28   |
> | HyTransformer              | 28.54   | 24.17  | 38.86   | 28.48 | 53.64  | 52.47   | 76.79   |
> | ARCI                       |  29.19   | 25.84  | 39.06   | 28.88 | 53.95  | 52.49   | 76.61   |
> | Model (w/ GAT)             | **30.43** | **26.25** | **40.80** | **30.18** | **56.23** | 52.93   | **76.97** |
> | Model (w/ HAT)             | 30.27   | 26.05  | 40.52   | 30.08   | 55.67   | **53.22** | 76.64   |

---

> ### Author Response · Authors · 2024-11-24
>
> Also to make the experimental section more comprehensive and extensive, we conduct additional experiment with two more clinical tasks of mortality prediction and readmission prediction. For each task we used all the baselines used in sequential diagnosis tasks plus relevant baselines in the following:
>
> [1] Nguyen, Phuoc, et al. "Deepr: a convolutional net for medical records (2016)." ArXiv160707519 Cs Stat (2016).
>
> [2] Bai, Shaojie, J. Zico Kolter, and Vladlen Koltun. "An empirical evaluation of generic convolutional and recurrent networks for sequence modeling." arXiv preprint arXiv:1803.01271 (2018).
>
> [3] Song, Lihong, et al. "Medical Concept Embedding with Multiple Ontological Representations." IJCAI. Vol. 19. 2019.
>
> [4] Choi, Edward, et al. "Learning the graphical structure of electronic health records with graph convolutional transformer." Proceedings of the AAAI conference on artificial intelligence. Vol. 34. No. 01. 2020.
>
> [5] Gao, Junyi, et al. "Stagenet: Stage-aware neural networks for health risk prediction." Proceedings of The Web Conference 2020. 2020.
>
> [6] Ma, Liantao, et al. "Adacare: Explainable clinical health status representation learning via scale-adaptive feature extraction and recalibration." Proceedings of the AAAI Conference on Artificial Intelligence. Vol. 34. No. 01. 2020.
>
> [7] Zhang, Chaohe, et al. "GRASP: generic framework for health status representation learning based on incorporating knowledge from similar patients." Proceedings of the AAAI conference on artificial intelligence. Vol. 35. No. 1. 2021.
>
> [8] Xu, Ran, et al. "Hypergraph transformers for ehr-based clinical predictions." AMIA Summits on Translational Science Proceedings 2023 (2023): 582.
>
> [9] Hadizadeh Moghaddam, Arya, et al. "Contrastive Learning on Medical Intents for Sequential Prescription Recommendation." Proceedings of the 33rd ACM International Conference on Information and Knowledge Management. 2024.
>
> ### Table A2: Prediction Performance for Hospital Readmission and Mortality
> | Model           | **PRAUC (Readmission)** | **ROCAUC (Readmission)** | **PRAUC (Mortality)** | **ROCAUC (Mortality)** |
> |-----------------|-----------------------------|------------------------------|---------------------------|----------------------------|
> | Transformer     | 66.22                      | 62.75                       | 10.17                    | 56.91                     |
> | retain          | 68.14                      | 63.99                       | 11.06                    | 57.66                     |
> | GCT [4]         | 68.16                      | 65.48                       | 10.48                    | 58.99                     |
> | TCN [2]         | 68.27                      | 64.28                       | 10.77                    | 57.78                     |
> | GRASP [7]       | 69.70                      | 65.24                       | 10.75                    | 58.72                     |
> | StageNet [5]    | 68.20                      | 65.22                       | 10.62                    | 57.79                     |
> | AdaCare [6]     | 68.73                      | 64.86                       | 11.00                    | 58.77                     |
> | Deepr [1]       | 69.63                      | 65.59                       | 11.18                    | 59.74                     |
> | GRAM            | 68.32                      | 64.36                       | 12.27                    | 58.50                     |
> | MMORE           | 68.13                      | 64.60                       | 12.37                    | 59.77                     |
> | KAME            | 67.89                      | 63.69                       | 12.15                    | 57.98                     |
> | HAP             | 67.82                      | 63.85                       | 11.15                    | 57.10                     |
> | ARCI [9]        | 68.03                      | 65.17                       | 11.93                    | 60.19                     |
> | HyTransformer [8]| 67.51                     | 63.30                       | 12.31                    | 57.63                     |
> | **OntoFAR**     | **70.41**                  | **66.15**                   | **14.39**                | **63.69**                 |

---

> > ### Comment · Reviewer_VcGN · 2024-12-03
> >
> > Thank you for your response

---

> > > ### Author Response · Authors · 2024-12-04
> > >
> > > Dear Reviewer VcGN,
> > >
> > > Thank you for updating your evaluation of our submission. We greatly appreciate your thoughtful feedback.
> > >
> > >
> > > We also completed the MIMIC4 experimentation for the additional tasks of mortality prediction and readmission prediction:
> > >
> > >
> > > | Model         | PRAUC (Readmission) | ROCAUC (Readmission) | PRAUC (Mortality) | ROCAUC (Mortality) |
> > > |---------------|---------------------|----------------------|-------------------|--------------------|
> > > | Transformer   | 63.08              | 62.45               | 3.05             | 60.22             |
> > > | retain        | 64.72              | 63.48               | 4.95             | 60.32             |
> > > | GCT           | 64.11              | 63.05               | 3.78             | 60.3              |
> > > | TCN           | 64.49              | 63.31               | 3.9              | 60.62             |
> > > | GRASP         | 66.08              | 64.78               | 4.21             | 57.44             |
> > > | StageNet      | 65.09              | 64.49               | 4.38             | 64.99             |
> > > | AdaCare       | 64.88              | 63.83               | 4.7              | 63.38             |
> > > | Deepr         | 67.23              | 65.01               | 3.7              | 64.79             |
> > > | GRAM          | 64.77              | 63.98               | 3.3              | 64.15             |
> > > | MMORE         | 65.02              | 63.87               | 4.01             | 62.24             |
> > > | KAME          | 64.11              | 63.31               | 3.2              | 61.11             |
> > > | HAP           | 65.4               | 64.1                | 3.83             | 63.03             |
> > > | ARCI          | 65.95              | 64.33               | 3.97             | 63.41             |
> > > | HyTransformer | 65.55              | 64.5                | 4.58             | 58.43             |
> > > | **OntoFAR**   | **68.17**          | **65.11**           | **8.23**         | **66.35**         |
> > >
> > >
> > > We did our best to fully address your concerns. If you have any additional comments or suggestions, we are more than happy to address them to ensure the quality of our submission. Thank you so much!

---

### Official Review · Reviewer_ZVQp · 2024-11-04

**Soundness:** 3
**Presentation:** 2
**Contribution:** 2
**Rating:** 5
**Confidence:** 5

**Summary:**

The paper introduces OntoFAR, a framework designed to enhance medical concept representations by leveraging multiple ontology graphs to enrich electronic health record (EHR) data. OntoFAR aims to address limitations in existing EHR models that often treat different ontologies (e.g., conditions, drugs) in isolation, thereby missing out on potentially valuable cross-ontology relationships.

This framework achieves improvement through a dual-directional message passing mechanism: Vertical Message Passing (VMP) within individual ontology hierarchies and Horizontal Message Passing (HMP) across co-occurring medical codes in EHR visits.

However, the proposed methodology was evaluated on a limited set of datasets and prediction tasks, showing only marginal performance gains. This raises concerns about its model generalizability. Also, the fact that it needs LLM (embeddings initialization) for training relatively small predictive models could raise the efficiency issues.

Additionally, the methodological approach lacks substantial novelty from a machine learning perspective. It primarily relies on modeling co-occurrences of codes within visits, which aligns closely with techniques already present in existing graph-based EHR predictive models, offering limited new contributions to the field.

**Strengths:**

1. Incorporation of Heterogeneous Ontology Motivation: The paper provides a strong motivation for utilizing heterogeneous ontologies, emphasizing the need for integrating multiple medical ontology systems to improve healthcare predictive models.

2. Demonstration of Model Strength through Comparative Experiments: Through extensive experiments comparing OntoFAR with existing methods, the paper demonstrates the model's superiority in predictive performance, particularly on tasks using real-world medical datasets like MIMIC-III and MIMIC-IV.

3. Evidence of Medical Ontology as a Key Feature in Predictive Models: The results indicate the significance of medical ontology in enhancing predictive accuracy, showing that it can serve as a critical feature in medical concept representation, effectively boosting the model's robustness and interpretability.

**Weaknesses:**

1. Proposed model novelty and contributions

(a) Effectiveness of LLM-based Initial Embedding: Results in Table 3 indicate that initializing embeddings with an LLM significantly improves performance. Further clarification is needed on how this initialization contributes to the overall performance of OntoFAR, as well as details on the LLM prompt design strategy. Additionally, it would be informative to assess the impact of initializing embeddings with Clinical-BERT, which is specifically trained on MIMIC medical concepts.

(b) VMP and HMP Design and Interpretation: VMP applies an established concept, and HMP seems to rely on a co-occurrence-based graph attention mechanism, which is a pre-existing technique. Although combining these two approaches appears to be a central contribution of the paper, it is unclear if HMP’s co-occurrence-based construction fully leverages inter heterogeneous medical ontologies. A more explicit discussion is needed on whether combining various ontologies in this way can genuinely contribute to model performance. (Author mentioned co-occurrence based model is utilization of "inter ontology".)

(c) Co-occurrence and Predictive Model: At the visit level, co-occurrence information may already be incorporated within the predictive model itself. If this is the case, it is unclear what additional benefits the medical concept encoder provides, even in Table 3 where HMP is highlighted. This rationale could benefit from further elaboration.

2. Limitations in Experiment Setup and Dataset Diversity

(a) Lack of Diversity in Predictive Models: Table 2 evaluates different medical concept encoders with transformer as the predictive model, yet same experiment results for RETAIN and TCN, which are shown in Table 1, are not included. Including similar results for RETAIN and TCN would provide a more comprehensive assessment of the model’s generalization capabilities.

(b) Limited Task Scope: This paper primarily focuses on a single task, sequential diagnosis prediction for the next visit. Expanding the evaluation to additional tasks would better demonstrate the generalizability and broader applicability of the proposed representations.

(c) Dataset Diversity: The experiments are conducted solely on the MIMIC dataset, which limits insights into the model’s robustness across datasets. Testing the model on additional datasets would strengthen evidence of its generalizability.

3. Model Evaluation and Performance

(a) Usage of Embeddings: Clarification is needed on whether the embeddings generated by the medical concept encoder are fixed or serve solely as initial embeddings. For example, GRAM, which serves as a baseline, uses an end-to-end approach with predictive model. Is OntoFAR primarily used to provide only initial embeddings for code representation?

(b) Marginal Improvement in Performance: The proposed model demonstrates only marginal performance gains, which makes it difficult to establish a clear advantage over existing approaches. This is especially evident when LLM embedding initialization is excluded, where the performance improvement seems negligible.

**Questions:**

1. Related Work: Additional discussion on how the proposed model distinctly differs from related studies would be beneficial. Clarifying the relationship between prior work and the proposed model would enhance the understanding of OntoFAR's unique contributions.

2. HGIP in Table 3: In Table 3, HGIP appears to exclude VMP. Does this mean that message passing was not applied at all, and were using the embeddings initialized by LLM used without further updates?

---

> ### Author Response · Authors · 2024-11-24
>
> Dear reviewer ZVQp,
>
> Thank you for your detailed and thoughtful review of our manuscript. We greatly appreciate your valuable feedback and the time you have taken to provide constructive suggestions. In the following response, we address your questions and present new experimental results to incorporate your recommendations and address your concerns.
>
> 1.	Issues of experiments.
> - **More recent baselines are necessary:** We added 2 more baselines to the task of sequential diagnosis prediction.
>
> [8] Xu, Ran, et al. "Hypergraph transformers for ehr-based clinical predictions." AMIA Summits on Translational Science Proceedings 2023 (2023): 582.
>
> [9] Hadizadeh Moghaddam, Arya, et al. "Contrastive Learning on Medical Intents for Sequential Prescription Recommendation." Proceedings of the 33rd ACM International Conference on Information and Knowledge Management. 2024.
> Please refer to Table A1 below.
>
> # Table A1: Sequential Diagnosis Prediction
> | Model                      | PR-AUC  | F1     | Acc@20  | 0-25%   | 25-50%  | 50-75%  | 75-100% |
> |----------------------------|---------|--------|---------|---------|---------|---------|---------|
> | Transformer                | 28.23   | 22.36  | 38.03   | 28.07   | 54.20   | 50.62   | 74.14   |
> | GRAM                       | 28.99   | 23.62  | 39.14   | 28.84   | 54.58   | 50.58   | 74.44   |
> | MMORE                      | 29.11   | 23.67  | 39.14   | 28.87   | 54.92   | 51.29   | 74.42   |
> | KAME                       | 28.52   | 23.18  | 38.36   | 28.19   | 55.13   | 50.09   | 73.79   |
> | HAP                        | 29.28   | 23.25  | 39.46   | 29.10   | 55.11   | 52.19   | 76.28   |
> | HyTransformer              | 28.54   | 24.17  | 38.86   | 28.48 | 53.64  | 52.47   | 76.79   |
> | ARCI                       |  29.19   | 25.84  | 39.06   | 28.88 | 53.95  | 52.49   | 76.61   |
> | Model (w/ GAT)             | **30.43** | **26.25** | **40.80** | **30.18** | **56.23** | 52.93   | **76.97** |
> | Model (w/ HAT)             | 30.27   | 26.05  | 40.52   | 30.08   | 55.67   | **53.22** | 76.64   |

---

> ### Author Response · Authors · 2024-11-24
>
> - **The proposed methodology was evaluated on a limited variety of prediction tasks**: We added two more clinical tasks: 1) mortality prediction and 2) hospital readmission prediction in additional to diagnosis prediction. Table A2 presents the results of these tasks alongside baseline comparisons, based on the MIMIC-III dataset. Experiments on the MIMIC-IV dataset are underway, and we will share the results once completed.
>
> In addition, we also added more relevant baselines for mortality prediction and readmission prediction:
>
> [1] Nguyen, Phuoc, et al. "Deepr: a convolutional net for medical records (2016)." ArXiv160707519 Cs Stat (2016).
>
> [2] Bai, Shaojie, J. Zico Kolter, and Vladlen Koltun. "An empirical evaluation of generic convolutional and recurrent networks for sequence modeling." arXiv preprint arXiv:1803.01271 (2018).
>
> [3] Song, Lihong, et al. "Medical Concept Embedding with Multiple Ontological Representations." IJCAI. Vol. 19. 2019.
>
> [4] Choi, Edward, et al. "Learning the graphical structure of electronic health records with graph convolutional transformer." Proceedings of the AAAI conference on artificial intelligence. Vol. 34. No. 01. 2020.
>
> [5] Gao, Junyi, et al. "Stagenet: Stage-aware neural networks for health risk prediction." Proceedings of The Web Conference 2020. 2020.
>
> [6] Ma, Liantao, et al. "Adacare: Explainable clinical health status representation learning via scale-adaptive feature extraction and recalibration." Proceedings of the AAAI Conference on Artificial Intelligence. Vol. 34. No. 01. 2020.
>
> [7] Zhang, Chaohe, et al. "GRASP: generic framework for health status representation learning based on incorporating knowledge from similar patients." Proceedings of the AAAI conference on artificial intelligence. Vol. 35. No. 1. 2021.
>
> [8] Xu, Ran, et al. "Hypergraph transformers for ehr-based clinical predictions." AMIA Summits on Translational Science Proceedings 2023 (2023): 582.
>
> [9] Hadizadeh Moghaddam, Arya, et al. "Contrastive Learning on Medical Intents for Sequential Prescription Recommendation." Proceedings of the 33rd ACM International Conference on Information and Knowledge Management. 2024.
>
> ### Table A2: Prediction Performance for Hospital Readmission and Mortality
> | Model           | **PRAUC (Readmission)** | **ROCAUC (Readmission)** | **PRAUC (Mortality)** | **ROCAUC (Mortality)** |
> |-----------------|-----------------------------|------------------------------|---------------------------|----------------------------|
> | Transformer     | 66.22                      | 62.75                       | 10.17                    | 56.91                     |
> | retain          | 68.14                      | 63.99                       | 11.06                    | 57.66                     |
> | GCT [4]         | 68.16                      | 65.48                       | 10.48                    | 58.99                     |
> | TCN [2]         | 68.27                      | 64.28                       | 10.77                    | 57.78                     |
> | GRASP [7]       | 69.70                      | 65.24                       | 10.75                    | 58.72                     |
> | StageNet [5]    | 68.20                      | 65.22                       | 10.62                    | 57.79                     |
> | AdaCare [6]     | 68.73                      | 64.86                       | 11.00                    | 58.77                     |
> | Deepr [1]       | 69.63                      | 65.59                       | 11.18                    | 59.74                     |
> | GRAM            | 68.32                      | 64.36                       | 12.27                    | 58.50                     |
> | MMORE           | 68.13                      | 64.60                       | 12.37                    | 59.77                     |
> | KAME            | 67.89                      | 63.69                       | 12.15                    | 57.98                     |
> | HAP             | 67.82                      | 63.85                       | 11.15                    | 57.10                     |
> | ARCI [9]        | 68.03                      | 65.17                       | 11.93                    | 60.19                     |
> | HyTransformer [8]| 67.51                     | 63.30                       | 12.31                    | 57.63                     |
> | **OntoFAR**     | **70.41**                  | **66.15**                   | **14.39**                | **63.69**                 |

---

> ### Author Response · Authors · 2024-11-24
>
> - **The paper uses two experimental datasets with the same scope**: While MIMIC-III and MIMIC-IV share some patient overlap, they differ significantly in critical aspects, making their joint use valuable. MIMIC-III covers ICU stays from 2001 to 2012 (46,000 patients), while MIMIC-IV spans 2008 to 2019 (383,220 patients) and includes newer coding systems like ICD-10-CM/PCS.
> Although eICU Collaborative Research Database is a another widely used public EHR dataset, the key features of medications in eICU are recorded using generic names rather than standardized ontology systems like RxNorm or ATC. Since our study focuses on integrating different ontologies for diagnoses, drugs, and procedures, eICU lacks an ontology system for each of these features.
> - **The performance gains are marginal.** This is especially evident when LLM embedding initialization is excluded: Our results are based on the complex task of predicting ICD-9 diagnosis codes for the next visit, involving 4,283 unique codes in MIMIC-III and 8,818 in MIMIC-IV. The large prediction space makes the task challenging, where the gains in proposed model represent significant improvements in predictive performance. In addition, the performances are obtained through a cross-validation setting.  Notably, our model excels in two key aspects: first, it performs significantly better at predicting less frequent (rare) diagnosis codes (Figure 3), and second, it demonstrates superior performance in data-limited settings (Figures 4 and 5).
> As shown in Table 3 (ablation study), initializing embeddings with LLM enhances overall performance. However, even without LLM, our model outperforms other baselines. Additionally, we have included new ablation experiments in Table A4 below, highlighting the contribution of each component across code frequency categories for your reference.

---

> ### Author Response · Authors · 2024-11-24
>
> 2.	**The methodological approach lacks substantial novelty from a machine learning perspective.**
> We believe our work is valuable to the field of EHR mining from the following perspectives.
> -	As existing approaches are limited by the isolation of different ontology systems (e.g., conditions vs. drugs), OntoFAR integrates diverse ontology graphs (diagnoses, prescriptions, drugs) and employs multifaceted graph learning across ontologies to enhance medical concept representation. To the best of our knowledge, this study is among the first to focus on fusing heterogeneous ontology systems.
> -	Our method jointly represents medical concepts across multiple ontology structures by performing message passing in two dimensions:
> •	Horizontal propagation uses hierarchical parallel attention graphs to link ontologies, capturing co-occurrence across granularities and integrating fine- and coarse-grained information for richer representation learning and EHR pattern mining.
> •	Vertical propagation within the ontology hierarchy facilitates intra-ontology concept association through the Hierarchical Graph Information Propagation. This two-round progressive encoding technique integrates information across all levels. It controls node update order to incorporate hierarchy and uses multi-head attention for multi-view representations, addressing EHR-ontology inconsistencies with a flexible, multi-sense approach.
> c.	OntoFAR leverages the large language models (LLMs) to understand each target concept with its ontology relationships, providing enhanced embedding initialization for concepts.
> d.	We conducted extensive experiments on two widely used EHR datasets, MIMIC-III and MIMIC-IV, focusing on sequential diagnosis prediction (plus the newly added tasks of readmission prediction and mortality prediction (in Table A2)). Our analysis includes performance enhancement when integrating EHR models (Figure 2), baseline comparisons (Table 2), ablation studies (Table 3), data insufficiency tests (Figures 4 and 5), and interpretative case studies (Figure 6).
> 3.	**Details and Clarification on the LLM prompt design strategy:**
> This is comprehensively explained in section 3.3. The key steps are summarized as follows:
> -	We use LLM to initialize the embeddings of medical concept codes for all ontologies (diagnosis, prescription, drugs) and all levels (child, parent, grandparent).
> -	We design a specific prompting strategy for each medical code based on code information (the concept names) and its ancestors’ in the ontology.
> -	We only extract dense embedding from LLM, we do not get any prediction or written inference from LLM. This approach avoids LLM hallucinations while still leveraging its valuable general knowledge.
> -	The LLM embeddings are generated and stored prior to the training phase.
> -	These embeddings will be used as initialization of medical concept embeddings (they are not freeze and will be tuned).
> -	Based on Table 3 in the paper (ablation study), This LLM embedding initialization improves the overall performance. (However, without LLM, our model still outperforms other baselines)
>
> 4.	**A more explicit discussion is needed on whether combining various ontologies in this way can genuinely contribute to model performance.**
> Integrating diverse ontologies lead to better medical concept representation and consequently better performance, for two reasons:
> -	Based on ablation study, Horizontal Massage Passing, which is the module connecting different ontologies, has the most contribution to the final results
> -	Compared to the baselines of GRAM, KAMME, MMORE, HAP, where ontology systems are separately treated, we perform significantly better. (Table 2 in the paper, and the Tables A1, A2 added above).
> We add more explicit discussion in the experimental section 5.2.

---

> ### Author Response · Authors · 2024-11-24
>
> 5.	**Related Work: Additional discussion on how the proposed model distinctly differs from related studies would be beneficial. Clarifying the relationship between prior work and the proposed model would enhance the understanding of OntoFAR's unique contributions.**
> We will add more discussion in related work section to explain our work’s differentiation. Clarifications are summarized as follows:
>
> Similarities:
> - Both our work and similar works (GRAM, KAME, MMORE, HAP ADORE, etc.) utilize the vertical hierarchy of medical ontology specifically leveraging the parent-child relationship.
> -	All these methods utilize some kind of attention-based method to capture this parent-child connection inside ontology.
>
> Distinction:
> - Our work is the first to combine diverse ontologies (diagnosis (ICD), drug (ATC), procedures (ICD))
> - We utilize the synergy of both vertical massage passaging (inside one ontology separately) and horizontal massage passaging (across multiple ontologies)
> - Our vertical massage passaging is more advanced and expressive. Refer to Hierarchical Graph Information Propagation (HGIP) section 3.5, which leverages a bottom-up sequence of graph attentions, propagating information throughout the hierarchy.
> -	We design a specialized prompt to generate LLM embeddings that initialize each node (medical concept) in our Meta Knowledge Graph.

---

> ### Comment · Reviewer_ZVQp · 2024-12-03
> **Still, I can't understand how the experiment results came out.**
>
> Thank you so much for your thorough and diligent responses.
>
> Many issues have been resolved thanks to your efforts, especially the extensive additional experiments you conducted, which I believe have significantly contributed to the robustness of the research.
>
> However, there are still some unresolved issues on my end.
>
> The aspect of leveraging heterogeneous ontology seems promising. It looks like applying HGIP in cases where various ontologies appear within the same visit in the EHR time-series dataset is a good approach.
>
> However, the limitation seems to lie in the fact that this method ultimately only initializes the embeddings of the code vocab that serve as inputs to the prediction model. (Why your proposed model couldn't utilize heterogenous ontologies with predictive model? - I mean, is there any other way for end-to-end learning to fully leverage HGIP with predictive model?)
>
> Therefore, I still do not fully understand what specific aspects of OntoFAR make it superior to other baselines, resulting in better performance. For instance, GRAM also leverages ontology, and its prediction model would learn whether codes appearing in different ontologies co-occur within the same visit.
>
> Given that OntoFAR utilizes heterogeneous ontology only for code embedding initialization, I am curious why GRAM, which employs end-to-end learning, performs worse. (I assumed, for a fair comparison, that GRAM also utilized codes from various ontologies in this paper experiment, just like OntoFAR.)
>
> Anyway, thank you for the good answer!
> I've raised the score.

---

> > ### Author Response · Authors · 2024-12-03
> > **Clarification: Our optimization approach is indeed end-to-end and task adaptive.**
> >
> > Dear Reviewer ZVQp,
> >
> > Thank you very much for appreciating our work and for taking the time to update your evaluation of our submission. We appreciate your thoughtful feedback. Below we provide you we additional clarification.
> >
> > **However, the limitation seems to lie in the fact that this method ultimately only initializes the embeddings of the code vocab that serve as inputs to the prediction model. (Why your proposed model couldn't utilize heterogenous ontologies with predictive model? - I mean, is there any other way for end-to-end learning to fully leverage HGIP with predictive model?)**
> >
> > Our model is a medical concept encoder, designed with plug-in capability for diverse predictive models. This means that, given various existing predictive models, we can integrate our ontology encoder to enhance medical concept representation learning and ultimately boost predictive performance.
> >
> > Meanwhile, our model is indeed trained in an **end-to-end** fashion like GRAM, which **does not** initialize medical concepts and pass them to downstream predictive models in separate stages. Instead, it integrates with the final predictive module and generates embeddings in a task-specific and final-model-specific manner. We train and validate the model on various predictive tasks: diagnosis, readmission, or mortality prediction.
> >
> > **Given that OntoFAR utilizes heterogeneous ontology only for code embedding initialization, I am curious why GRAM, which employs end-to-end learning, performs worse. (I assumed, for a fair comparison, that GRAM also utilized codes from various ontologies in this paper experiment, just like OntoFAR.)**
> >
> > Yes, your assumption is correct. In the implementation of the baselines, we incorporate various ontologies. The results for GRAM were obtained using all medical codes (diagnosis, prescriptions, and drugs with different ontologies). However, a limitation of GRAM is that the representation learning of concepts of different ontologies does not talk with each other in an integrated way (as how we facilitate the ontology fusion in horizontal message passing).
> >
> > In GRAM, only the vertical information propagation is **independently** conducted inside each ontology graph. This pure vertical propagation in GRAM incorporates ancestor concepts of a code within **a single** ontology, however, it does not account for relationships across **multiple ontologies** (e.g., the ancestor concepts in drug ontology for a diagnosis code). To address this gap, we integrate both horizontal message passing (to consider interactions across multiple ontologies) and vertical message passing (to capture code interactions within individual ontology), to facilitate comprehensive multi-ontology representation learning.

---

> ### Author Response · Authors · 2024-12-03
> **Additional Experimentation for New tasks of Mortality Prediction and Readmission Prediction with MIMIC-IV dataset.**
>
> As we promised in our earlier response, we also completed the MIMIC-IV experimentation for the additional tasks of mortality prediction and readmission prediction (results are percentage %):
>
> | Model         | PRAUC (Readmission) | ROCAUC (Readmission) | PRAUC (Mortality) | ROCAUC (Mortality) |
> |---------------|---------------------|----------------------|-------------------|--------------------|
> | Transformer   | 63.08              | 62.45               | 3.05             | 60.22             |
> | retain        | 64.72              | 63.48               | 4.95             | 60.32             |
> | GCT           | 64.11              | 63.05               | 3.78             | 60.3              |
> | TCN           | 64.49              | 63.31               | 3.9              | 60.62             |
> | GRASP         | 66.08              | 64.78               | 4.21             | 57.44             |
> | StageNet      | 65.09              | 64.49               | 4.38             | 64.99             |
> | AdaCare       | 64.88              | 63.83               | 4.7              | 63.38             |
> | Deepr         | 67.23              | 65.01               | 3.7              | 64.79             |
> | GRAM          | 64.77              | 63.98               | 3.3              | 64.15             |
> | MMORE         | 65.02              | 63.87               | 4.01             | 62.24             |
> | KAME          | 64.11              | 63.31               | 3.2              | 61.11             |
> | HAP           | 65.4               | 64.1                | 3.83             | 63.03             |
> | ARCI          | 65.95              | 64.33               | 3.97             | 63.41             |
> | HyTransformer | 65.55              | 64.5                | 4.58             | 58.43             |
> | **OntoFAR**   | **68.17**          | **65.11**           | **8.23**         | **66.35**         |
>
>
> We sincerely appreciate your thoughtful feedback and have made every effort to thoroughly address your concerns. If you have any additional comments or suggestions, we would be more than happy to incorporate them to further enhance the quality of our submission. Thank you once again for your time and valuable insights!

---

### Meta-Review · Area_Chair_fMN8 · 2024-12-21

**Metareview:**

This paper introduces OntoFAR, a framework integrating multiple ontologies with vertical and horizontal message passing, enhancing medical concept representation and predictive performance in EHR models across various datasets and tasks.

Strengths
- OntoFAR demonstrates consistent performance improvements across multiple predictive tasks (diagnosis, mortality, readmission) on datasets like MIMIC-III and MIMIC-IV
- Authors showed dedication in addressing reviewer concerns by adding new tasks, baseline comparisons, and detailed analyses.

Weaknesses
- OntoFAR's performance heavily depends on LLM-based embedding initialization, with marginal improvements over baselines when LLMs are excluded, raising fairness and novelty concerns
- In the experiments, model comparisons lack some relevant recent methods (e.g., KAMPNet, SeqCare), undermining the paper’s claim of superiority.
- The dual-dimensional message passing approach can be seen as a combination of existing methods (e.g., co-occurrence-based models), reducing the paper’s methodological contribution.

**Additional Comments On Reviewer Discussion:**

During the rebuttal, the authors made significant effort to address as many reviewer concerns as possible. However, several issues persist even after such effort such as lack of most relevant baselines (e.g. KAMPNet and SeqCare), diminished performance improvement over the baselines when the LLM initialization is removed, and lack of originality of the dual-dimension message passing.

---

### Decision · Program_Chairs · 2025-01-22

Reject